# Distal axotomy enhances retrograde presynaptic excitability onto injured pyramidal neurons via trans-synaptic signaling

Tharkika Nagendran[1,2], Rylan S. Larsen [2,3,8], Rebecca L. Bigler [4], Shawn B. Frost[5,6], Benjamin D. Philpot[2,3,7], Randolph J. Nudo[5,6] & Anne Marion Taylor [1,2,7]

Injury of CNS nerve tracts remodels circuitry through dendritic spine loss and hyper-excitability, thus influencing recovery. Due to the complexity of the CNS, a mechanistic understanding of injury-induced synaptic remodeling remains unclear. Using microfluidic chambers to separate and injure distal axons, we show that axotomy causes retrograde dendritic spine loss at directly injured pyramidal neurons followed by retrograde presynaptic hyper-excitability. These remodeling events require activity at the site of injury, axon-to-soma signaling, and transcription. Similarly, directly injured corticospinal neurons in vivo also exhibit a specific increase in spiking following axon injury. Axotomy-induced hyper-excitability of cultured neurons coincides with elimination of inhibitory inputs onto injured neurons, including those formed onto dendritic spines. *Netrin-1* downregulation occurs following axon injury and exogenous netrin-1 applied after injury normalizes spine density, presynaptic excitability, and inhibitory inputs at injured neurons. Our findings show that intrinsic signaling within damaged neurons regulates synaptic remodeling and involves netrin-1 signaling.

[1] UNC/NCSU Joint Department of Biomedical Engineering, UNC-Chapel Hill, Campus box 7575, Chapel Hill, NC 27599-7575, USA. [2] UNC Neuroscience Center, UNC-Chapel Hill, Campus box 7250, Chapel Hill, NC 27599-7250, USA. [3] Department of Cell Biology and Physiology, UNC-Chapel Hill, Campus box 7545, Chapel Hill, NC 27599-7545, USA. [4] Curriculum in Genetics and Molecular Biology, UNC-Chapel Hill, Chapel Hill, NC 27599, USA. [5] Landon Center On Aging, University of Kansas Medical Center, 3901 Rainbow Blvd., Kansas City, KS 66160, USA. [6] Department of Rehabilitation Medicine, University of Kansas Medical Center, 3901 Rainbow Blvd., Kansas City, KS 66160, USA. [7] Carolina Institute for Developmental Disabilities, Campus box 7255, Chapel Hill, NC 27599-7255, USA. [8] Present address: Allen Institute for Brain Science, 615 Westlake Avenue North, Seattle, WA 98109, USA. Correspondence and requests for materials should be addressed to A.M.T. (email: anne.marion.taylor@gmail.com)

Acquired brain injuries, such as occur in stroke and traumatic brain injury, induce significant synaptic reorganization, even in uninjured cortical regions remote from the site of damage[1–3]. This enhanced neural plasticity supports formation of new connections and expansion of cortical territories, well-described in humans using neuroimaging and non-invasive stimulation techniques[1, 2, 4, 5]. However, the cellular mechanisms of this injury-induced plasticity remain largely unknown.

In healthy brains, long projection neurons with somatodendritic domains housed in cerebral cortex extend axons into numerous distant areas of the central nervous system (CNS), including the spinal cord and the apposing cortical hemisphere. When these remote areas are injured, long projection axons are damaged and injury signals propagate retrogradely to somatodendritic domains. Retrograde injury signal propagation leads to somatic responses such as chromatolysis and new transcription[6, 7]. For example, after damage to corticospinal axons resulting from spinal cord injury, dendritic spines in motor cortex undergo time-dependent changes in morphology including decreased spine density and alterations in spine length and diameter[8]. Loss of local inhibition also occurs at somatodendritic regions following injury, which is thought to unmask preexisting excitatory connections and result in enhanced

excitability[2, 9, 10]. These findings suggest that a cascade of events occurs following distal axonal injury involving retrograde axon-to-soma signaling and then trans-synaptic signaling from the injured neuron to uninjured presynaptic neurons causing synaptic changes and enhanced excitability.

Due to the heterogeneity and complexity of the CNS, intrinsic neuronal responses to distal axon injury and their contributions to synaptic remodeling remain unclear. Reduced preparations are necessary for examining neuron-specific responses and provide a more experimentally tractable model system to identify and screen drugs to improve neuronal function following injury. Microfluidic chambers are useful for compartmentalization of many types of neurons, including cortical and hippocampal neurons, and allow axons to be injured and manipulated without physically disturbing the proximal neurons housed within the chamber's somatodendritic compartment[11–13]. Because brain injury and disease preferentially affect long projection pyramidal neurons within the CNS[14, 15], we sought to determine the progression of events that occur intrinsically in these neurons following distal axotomy that lead to synaptic remodeling.

Here we show that axotomized pyramidal neurons undergo dendritic spine loss followed by a trans-synaptic enhancement in

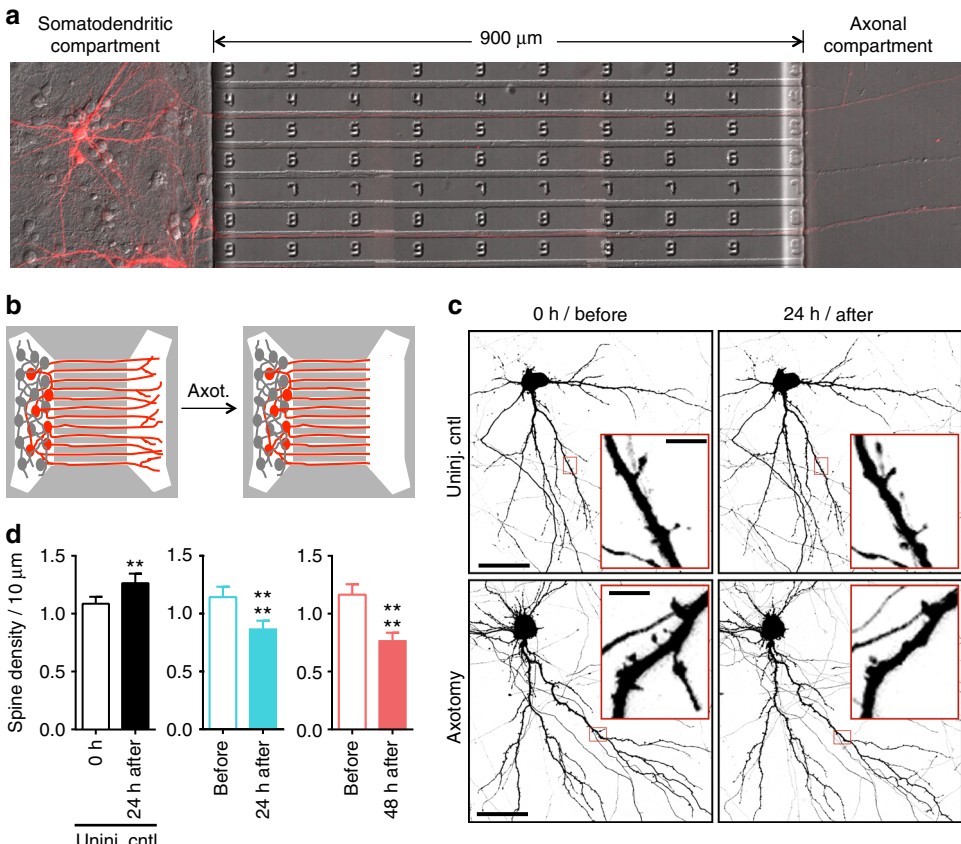

**Fig. 1** Dendritic spine density of pyramidal neurons within microfluidic chambers before and after axotomy. **a** 14 DIV rat hippocampal neurons cultured within a microfluidic chamber. Pyramidal neurons were retrogradely labeled using a G-deleted mCherry rabies virus added exclusively to the axonal compartment. **b** Cartoon illustration of in vitro axotomy (Axot.) within microfluidic chambers to selectively axotomize a subset of labeled neurons (*red*) that extend axons through the microgroove region. Axons of neighboring uncut neurons (*gray*) remain within the somatodendritic compartment. **c** Representative images of mCherry-labeled neurons and dendritic segments (inverted fluorescence) from repeated live imaging of uninjured control chambers (Uninj. cntl) and axotomized chambers. Axotomized chambers were imaged before axotomy on 13 DIV (before) and uninjured control chambers were also imaged on 13 DIV (0 h); both conditions were then imaged at 14 DIV (24 h after). Image and inset *scale bars*, 50 and 5 μm, respectively. **d** Quantification of spine density illustrated in **c**. *Uninj. cntl*: n = 20 dendrites; 5 neurons; 3 chambers over 3 experiments; #spines/TDL: 315/2698 μm (0 h), 388/2737 μm (24 h after). *Axot.*: n = 26 dendrites; 5 neurons; #spines/TDL: 405/3569 μm (before), 274/2998 μm (24 h after), and 229/2821 μm (48 h after). 3 chambers over 3 experiments. TDL: total dendritic length. Paired two-tailed *t*-test. **p < 0.01, ****p < 0.0001. *Error bars*, s.e.m

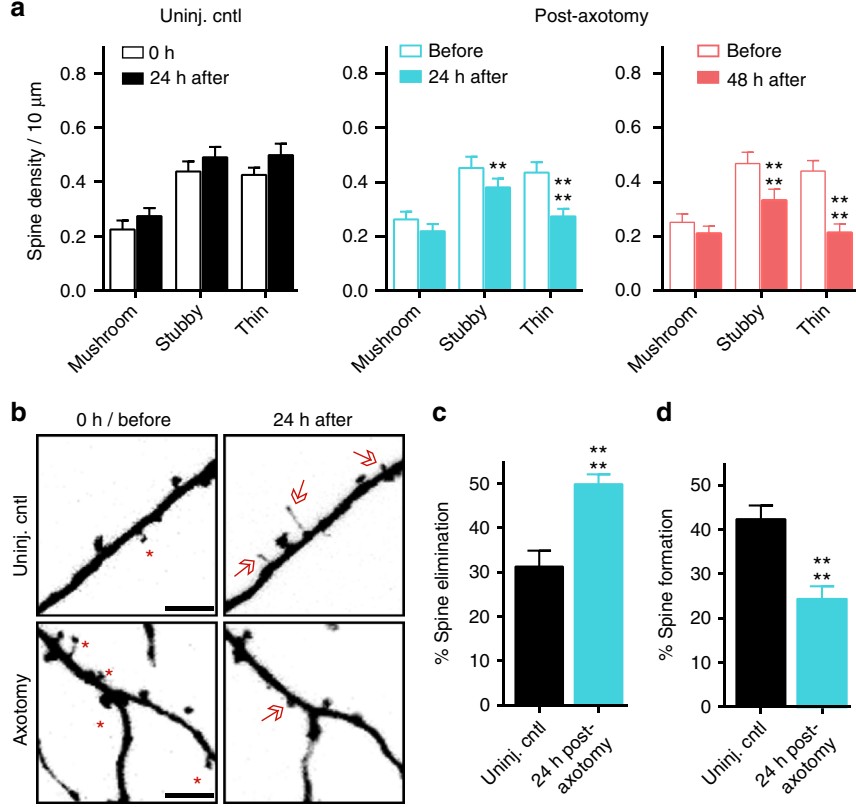

**Fig. 2** Analysis of spine types lost and percentages of spines formed and eliminated after axotomy. **a** Quantification of stubby, mushroom, and thin spine densities before axotomy, 24 h post-axotomy, 48 h post-axotomy, and in uninjured control chambers. Uninjured controls and axotomized samples were imaged beginning at 13 DIV (labeled "0 h" or "before", respectively). Uninj. cntl: $n = 20$ dendrites; 5 neurons; 3 chambers over 3 experiments; #spines/TDL: 315/2698 μm (0 h), 388/2737 μm (24 h after). Axot.: $n = 26$ dendrites; 5 neurons; #spines/TDL: 405/3569 μm (before), 274/2998 μm (24 h after), and 229/2821 μm (48 h after). 3 chambers over 3 experiments. Repeated-measure two-way analysis of variance (ANOVA), Bonferroni post hoc test. **b** Representative images of dendritic segments from uninjured control and axotomized chambers at 0 h or before axotomy, respectively, and 24 h after. *Asterisks* indicate spines eliminated; *arrows* indicate formation of new spines. *Scale bars*, 5 μm. **c**, **d** Bar graphs represent percentage of spines eliminated **c** and newly formed **d** 24 h post-axotomy and in age-matched uninjured control chambers. Uninj. cntl: $n = 28$ dendrites; 6 neurons; #spines eliminated: 92; #spines formed: 177; TDL (0 h, 24 h after): 3147, 3148 μm. Axot: $n = 35$ dendrites; 6 neurons; #spines eliminated: 208; #spines formed: 75; TDL (before, 24 h after): 4290, 3865 μm. 3 chambers over 3 experiments. Unpaired two-tailed *t*-test. **$p < 0.01$, ****$p < 0.0001$. *Error bars*, s.e.m

presynaptic excitability. We find that directly injured neurons preferentially exhibit enhanced excitability, which coincides with the loss of inhibitory inputs onto injured neurons. Our evidence suggests that these synaptic remodeling events require retrograde signaling from the site of axon injury to the nucleus to rapidly activate transcription. Netrin-1 is significantly downregulated following axotomy and the application of exogenous netrin-1 protein several hours after axotomy restores spine density and normalizes presynaptic excitability, including the fraction of inhibitory inputs onto injured neurons.

## Results

**In vitro model to study axon injury of pyramidal neurons.** To investigate how distal axon injury remodels synapses contacting injured neurons, we used a microfluidic approach to compartmentalize cultured neurons. Microfluidic chambers containing microgroove-embedded barriers ~900 μm in length were used to compartmentalize axons independently from dendrites and somata of rat central neurons as demonstrated previously (Supplementary Fig. 1a)[11, 12, 16]. Using this approach we subjected neurons to distal axotomy ~1 mm away from their physically undisturbed dendrites and somata[11, 16]. We used hippocampal neurons harvested from embryonic rats to generate a

more consistent, enriched population of pyramidal neurons (85–90% pyramidal) compared with similarly harvested cortical neurons. Further, hippocampal neurons exhibit morphology characteristic of maturing pyramidal neurons in vivo[17]; the remaining hippocampal neurons are mostly inhibitory GABAergic interneurons[18]. To identify neurons with axons projecting into the axonal compartment, we retrogradely labeled neurons by applying a G-deleted rabies virus expressing fluorescent proteins (incompetent for trans-synaptic transfer) to the axonal compartment and characterized the morphology of the labeled neurons. We found that 94% (42 of 45) of virally labeled neurons were pyramidal neurons and the remaining were unclassifiable (Fig. 1a). When these neurons were cultured within the microfluidic chamber and axotomized within the axonal compartment (Fig. 1b), there was no loss in viability post-axotomy (Supplementary Fig. 1), similar to in vivo findings[19], and injured axons regrew[11, 16]. Supporting the use of this approach, we previously found that axotomy performed within the microfluidic chambers induced rapid expression of the immediate early gene *c-fos*[11], as reported in vivo[20]. Further, neurons labeled with retrograde tracer, Alexa 568-conjugated cholera toxin, showed a significant decrease in Nissl staining in the somata reflective of chromatolysis at 24 h post-axotomy[21] (Supplementary Fig. 1). Together, this model recapitulated key features of axotomy in vivo.

**Spine density decreases after distal axon injury**. Decreased spine density is seen in vivo in models of traumatic brain injury and spinal cord injury[22, 23]. To determine whether similar structural changes occur in cultured pyramidal neurons following distal axotomy, we quantified spine density within the somatodendritic compartment of axotomized neurons that were retrogradely labeled using mCherry rabies virus. Spine density significantly declined 24 h and 48 h post-axotomy compared to before axotomy (Fig. 1c, d). In contrast, uninjured control neurons showed increased spine density as expected to occur during normal maturation (Fig. 1c, d).

We next analyzed specific spine types that were lost. We found a preferential loss in the density of thin and stubby spines at both 24 h and 48 h post-axotomy compared to pre-axotomy (Fig. 2a). The density of mushroom spines remained stable at both 24 h and 48 h after axotomy, unlike in the uninjured control neurons where spine density of all spine types increased. The reduction in spine density following axotomy suggests that either dendritic spines were being eliminated or, conversely, that there was a reduction in new spine formation following axotomy. Further analysis of our before and after axotomy images revealed that axotomy caused both a significant

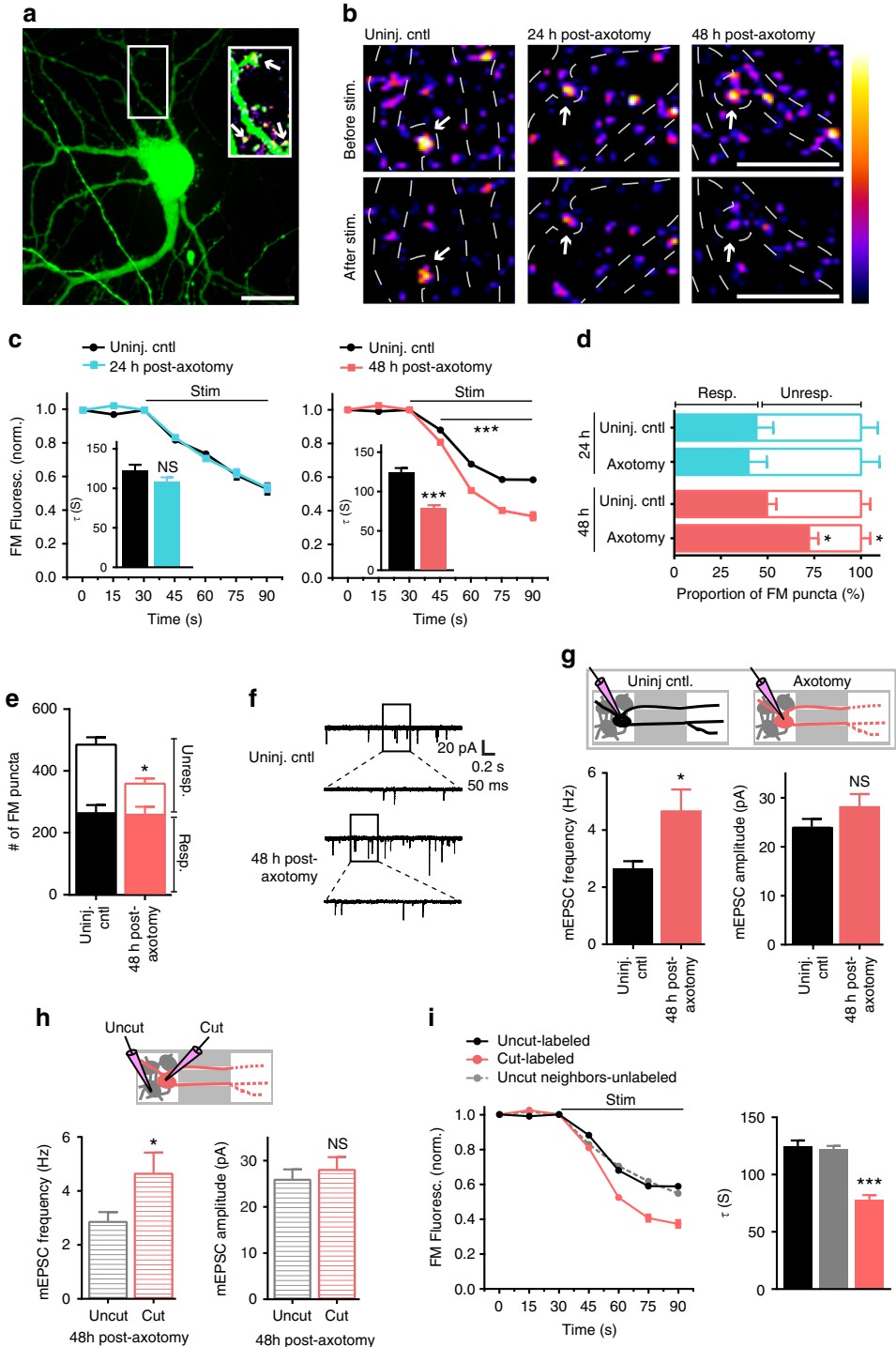

increase in the percentage of spines eliminated and a significant reduction in the percentage of new spines formed 24 h post-axotomy (Fig. 2b–d). Thus, axotomy affected both elimination and formation of spines to result in lower dendritic spine density.

**Increased synaptic vesicle release rate follows axon injury**. To further evaluate how synapses are modified following distal axon injury, we next investigated whether presynaptic release properties were altered at synapses onto injured neurons. To address this question, we retrogradely infected neurons using a modified enhanced green fluorescent protein (eGFP) rabies virus to label injured neurons and then used FM dyes to optically measure synaptic vesicle release onto these directly injured neurons (Fig. 3a). The use of FM dyes provided us with an unbiased method to label a majority of presynaptic terminals within the somatodendritic compartment[24]. FM puncta highly colocalized with synapsin1 immunolabeling (93%), which is present at both inhibitory and excitatory terminals, validating our FM dye loading strategy (Supplementary Fig. 2). We examined the synaptic vesicle release rate of FM puncta that colocalized with axotomized eGFP expressing neurons. At 24 h post-axotomy, there was no change in synaptic vesicle release rate compared to eGFP expressing uninjured control samples (Fig. 3b, c). In contrast, 48 h after axotomy synaptic vesicle release rate was significantly enhanced (Fig. 3c). Further, the FM decay time constant, $\tau$, which has been inversely correlated with release probability[25] was significantly reduced at 48 h post-axotomy (control: 124.8 s $\pm$ 5.487 vs.

axotomy: 78.65 s $\pm$ 3.922; $p < 0.0001$). These results were similar to those obtained by examining the entire image field of FM puncta closest to the barrier region within the somatodendritic compartment where a large percentage of axotomized neurons reside (Supplementary Fig. 3). The difference in presynaptic release rate persisted, though modestly, at 4 day post-axotomy in these cultured neurons (control: 95.14 s $\pm$ 1.282 vs. axotomy: 77.19 s $\pm$ 1.165; $p < 0.0001$; Supplementary Fig. 3). Together, these data suggest a delayed and persistent increase in synaptic vesicle release rate that occured following dendritic spine loss.

Next, we performed two control experiments to determine (1) whether cortical cultures, which have more neuron variaiblity than hippocampal cultures, would behave similarly to hippocampal cultures used in our experimental model, and (2) whether axotomy of axons forming synapses onto postsynaptic targets would yield similar effects as axotomy of untargeted axons. First, we performed the FM unloading experiments with cortical

neurons harvested from embryonic rats and found that these cultures showed similar changes in presynaptic release 48 h post-axotomy (Supplementary Fig. 3). To address the second question, we added a small number of target neurons to the axonal compartment during cell plating. We previously demonstrated that synapses form between two neuron populations plated into opposing compartments[12]. Axotomy of this targeted population of neurons resulted in similar changes in presynaptic release rate as axotomy of untargeted axons (Supplementary Fig. 3).

Dendritic spine density is lower following injury, suggesting fewer synapses, thus, we next wondered whether the balance of responsive to unresponsive presynaptic terminals might be altered following axotomy to account for the enhancement in excitability. We measured the proportion of FM puncta that unloaded (responsive) or did not unload (unresponsive) in response to field stimulation using extracellular electrodes[24] (Fig. 3d). At 24 h post-axotomy when spine density was decreased, we observed no change in the fraction of responsive and unresponsive FM puncta compared to uninjured controls (Fig. 3d). However at 48 h post-axotomy, a significantly larger proportion of puncta were responsive compared to puncta within uninjured control chambers (Fig. 3d). Further, at 48 h post-axotomy we found an overall decrease in the number of loaded FM puncta (Fig. 3e). Together, our data suggest that distal axon injury leads to an overall decrease in synapses, including the number of presynaptic terminals, but that the smaller number of presynaptic terminals is more responsive to stimulation.

**Enhanced glutamate release at synapses onto injured neurons**. Our results support that distal axotomy triggers a retrograde and trans-synaptic cascade of events leading to enhanced neurotransmitter release rate. To confirm this, we performed electrophysiological recordings of $\alpha$-amino-3-hydroxy-5-methyl-4-isoxazolepropionic acid receptor (AMPAR)-mediated miniature excitatory postsynaptic currents (mEPSCs) from axotomized neurons 48 h post-axotomy and their age-matched uninjured controls. Biocytin was used to fill neurons following each recording to determine whether neurons extended axons into the axonal compartment and were axotomized. Axotomized neurons had a significant increase in mEPSC frequency, confirming our FM data and supporting an increased rate of presynaptic glutamate release (Fig. 3f, g). Membrane properties were equivalent between axotomized and uninjured control neurons, demonstrating that the health of these axotomized neurons was not substantially compromised (Supplementary Table 1). We

**Fig. 3** Presynaptic excitability at synapses onto axotomized neurons. **a** A representative neuron retrogradely labeled with a modified eGFP rabies virus via the axonal compartment. Enlarged region shows FM puncta colocalized with eGFP dendrites and spines (*arrows*). ImageJ 'fire' color look-up-table for FM puncta shown in the next panel. *Scale bar*, 20 μm. **b** Representative images show FM puncta colocalized with eGFP dendrites (outlined in *white dashed lines*) before and after field stimulation in uninjured control, and 24 h and 48 h post-axotomy. *Arrows* highlight destaining at spines. *Scale bars*, 10 μm. **c** FM unloading curves of colocalized puncta 24 h post-axotomy (control, $n = 185$ puncta; axotomy, $n = 256$ puncta) and 48 h post-axotomy (control, $n = 232$ puncta; axotomy, $n = 322$ puncta). Two-way ANOVA, Bonferroni post hoc test. Inset shows FM decay time constant ($\tau$) for puncta with $\tau < 360$ s (24 h control, $n = 151$; 24 h axotomy, $n = 201$; 48 h control, $n = 211$; 48 h axotomy, $n = 304$). **b, c** Unpaired two-tailed $t$-test. Each condition includes 5–6 chambers/neurons over 3 experiments. **d** Percent responsive and unresponsive FM puncta per neuron field ($n = 8$ fields/chambers; 4 experiments). Unpaired two-tailed $t$-test, axotomy vs. control for each time point. **e** Number of responsive and unresponsive FM puncta per frame at 48 h post-axotomy ($n = 11$ chambers; 5 experiments). Unpaired two-tailed $t$-test, for unresponsive puncta. **f** Representative mEPSC traces 48 h post-axotomy. **g** mEPSC frequency and amplitude at 48 h post-axotomy (control, $n = 17$ neurons; axotomy, $n = 20$ neurons; 4 experiments). *Inset*: cartoon depicts recordings from either uninjured control neurons (*black*) or directly injured neurons (*red*). **h** Analysis of mEPSC frequency and amplitude of cut neurons [cut (*red*), $n = 10$ neurons] compared to neighboring uncut neurons within axotomized chambers [uncut (*gray*), $n = 10$ neurons]. **g, h** Unpaired two-tailed $t$-test, Welch's correction. **i** FM unloading of neighboring uncut neurons identified by lack of eGFP (uncut neighbors, $n = 816$ puncta), uninjured control neurons (uncut-labeled, $n = 232$), and axotomized labeled neurons (cut-labeled, $n = 322$). Two-way ANOVA, Bonferroni post hoc test; each condition, 5 chambers and 3 experiments. Decay time constant ($\tau$) of FM puncta at 48 h post-axotomy (uncut-labeled, $n = 211$; cut-labeled, $n = 304$; and uncut neighbors unlabeled, $n = 703$). One-way ANOVA, Bonferroni post hoc test. *$p < 0.05$, ***$p < 0.001$. *Error bars*, s.e.m

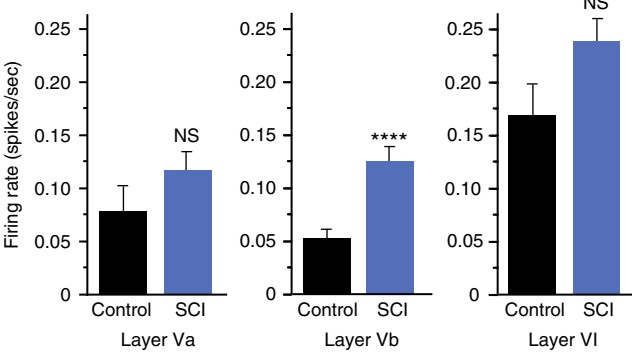

**Fig. 4** Spontaneous single-unit firing rate in hindlimb cortex 4–18 weeks following spinal cord injury. Mean spontaneous firing rates of isolated single-units in layers Va, Vb, and VI of the hindlimb motor cortex in control rats ($n = 5$) and rats with a spinal cord contusion at T9-10 ($n = 14$). Laminar estimates are based on the depths of electrode sites on a single-shank multi-electrode array relative to the cortical surface[26]. In each rat, single-unit (spike) activity was sampled from 4 to 6 locations within neurophysiologically identified hindlimb motor cortex. Data represent the mean firing rates of 1744 isolated units in Layers Va (control, $n = 124$; SCI, $n = 312$), Vb (control, $n = 155$; SCI, $n = 390$), and VI (control, $n = 217$; SCI, $n = 546$). Two-tailed t-test, $t = 3.99$, ****$p < 0.0001$

observed a trend towards an increase in mEPSC amplitude following axotomy, however this effect was not significant (Fig. 3g).

We next wondered if the increased spontaneous release rate of glutamate was specific to directly injured neurons or more globally affected neighboring, uncut neurons. To address this, we quantified mEPSC frequency between uncut and cut neurons within the same axotomized chamber. In recordings from directly injured neurons, axotomy specifically increased mEPSC frequency. However, neighboring uncut neurons that did not extend axons into the axonal compartment, did not have an increased mEPSC frequency (Fig. 3h). To further examine the effects of direct injury to axotomized neurons, we quantified FM release rate at nearby uninjured neurons that were not infected with the retrograde eGFP rabies virus. We found that the release rate was significantly decreased at these locations compared with synapses on directly axotomized neurons and not significantly different than at control uninjured neurons labeled with eGFP rabies virus (Fig. 3i). These observations confirmed that axotomy altered glutamatergic synaptic input onto injured neurons. Further, directly injured neurons trans-synaptically influenced presynaptic glutamate release without affecting nearby synapses at uninjured neurons.

**SCI induces persistent and enhanced firing in layer Vb.** To evaluate the in vivo relevance of our findings, we sought to determine whether distal injury of long projection neurons in vivo would preferentially induce enhanced excitability in these injured neurons. To do this, we wanted to use an in vivo model in which axonal damage occurs far from somata to minimize other effects of injury (e.g., inflammation and metabolic changes). We used a rat SCI model described previously[26] in which animals were subjected to a spinal cord contusion injury at thoracic level T9-10, and recording electrodes were implanted into the neurophysiologically identified hindlimb motor cortex in ketamine-anesthetized animals. Electrode sites on single-shank microelectrode arrays (Neuronexus, Ann Arbor, MI, USA) extended through cortical layers V and VI, allowing simultaneous

recording throughout these cortical layers. Effective injury to the corticospinal neurons innervating hindlimb motor neuron pools in the spinal cord was confirmed by stimulating electrode sites and confirming loss of evoked hindlimb movement. At each cortical location, 5 min of neural data was collected for offline analysis. At the end of the procedure, neural spikes were discriminated using principle component analysis. We examined firing rates[27] within layers Va, Vb, and VI between 4 weeks and 18 weeks post-SCI and compared the data to sham control animals. We found that the firing rate within layer Vb was significantly increased after SCI compared to sham controls (Fig. 4). Layer Vb contains the highest density of corticospinal somata, with estimates of nearly 80% of large pyramidal cells[28]. Also, after spinal cord injury, chromatolytic changes occur preferentially in layer Vb[29]. In layers Va and VI, which have few (layer Va) or no (layer VI) corticospinal neurons, we found that firing rates were not statistically different between SCI animals and sham controls. Together, these data confirm a persistent increase in spontaneous firing rates in remotely injured corticospinal neurons, and support the relevance of our in vitro model system.

**Axotomy eliminates inhibitory terminals onto injured neurons.** Loss of inhibition following distal injury contributes to enhance excitability in vivo, thus we wanted to test whether axotomy in our culture system results in a similar loss of inhibitory terminals that release γ-aminobutyric acid (GABA). We performed retrospective immunostaining to determine the fraction of vGLUT1 or GAD67-positive FM puncta at 48 h post-axotomy (Fig. 5a, b). We found that axotomy did not alter the fraction of glutamatergic terminals, but significantly diminished the fraction of GAD67-positive puncta within the somatodendritic compartment. Further, we examined the fraction of vGLUT1 or vGAT puncta colocalized with axotomized neurons labeled with an eGFP rabies virus (Fig. 5c). These results confirmed the preferential absence of inhibitory terminals following axotomy while the fraction of vGLUT1 puncta remained equivalent to uninjured control neurons.

To determine whether inhibitory synapses were functionally altered following axotomy, we recorded miniature inhibitory postsynaptic currents (mIPSCs) from axotomized and uninjured chambers 48 h post-axotomy (Fig. 5d–f). We found that mIPSCs were more frequent in axotomized cultures compared with uninjured neurons, suggesting that while there are fewer inhibitory terminals, the remaining terminals have an increased rate of spontaneous GABA release. We next asked whether this change in inhibitory synapse function was restricted to directly injured neurons. Within the axotomized cultures, we compared both cut and uncut neurons and found that the mIPSC frequency was increased in both groups, but was not different between the directly axotomized neurons and their uncut neighbors. This suggests that the alteration of inhibitory synaptic transmission following axotomy affects both directly injured and neighboring, uninjured neurons.

Although the majority of GABAergic synapses are found on dendritic shafts or cell bodies, a minor population is also found on dendritic spines[30, 31] (Supplementary Fig. 4). Inhibitory synapses formed on dendritic spines allow for compartmentalization of dendritic calcium levels involved in regulation of neuronal activity[32, 33]. To investigate whether dendritic spines receiving inhibitory inputs (i.e., inhibited spines) are lost following axotomy, we quantified the number of inhibitory and excitatory presynaptic terminals onto spines of cultured pyramidal neurons using retrospective immunostaining for inhibitory (vGAT) and excitatory (vGLUT1) synapse markers. We found a significant

decrease in the fraction of vGAT-positive spines at 48 h post-axotomy compared to uninjured control (Fig. 5g–i) with no significant influence on glutamatergic spines. Together, our data suggest that axotomy caused a preferential loss of inhibitory terminals onto axotomized neurons, including inhibitory terminals formed onto dendritic spines, and that increased spontaneous GABAergic transmission might compensate to some degree for these lost terminals.

**Local activity and transcription regulate remodeling.** Efficient axon regeneration requires signaling from the site of injury to the nucleus in multiple model systems[6], yet the signaling events required for synaptic remodeling following distal axotomy remain unclear. Breach of the axonal membrane following axon injury causes an influx of calcium and sodium ions into the intra-axonal space, potentially influencing signaling to the nucleus and gene expression. To determine whether local influx of sodium and calcium ions at the time of injury is required for axotomy-induced spine loss, we performed axotomy within

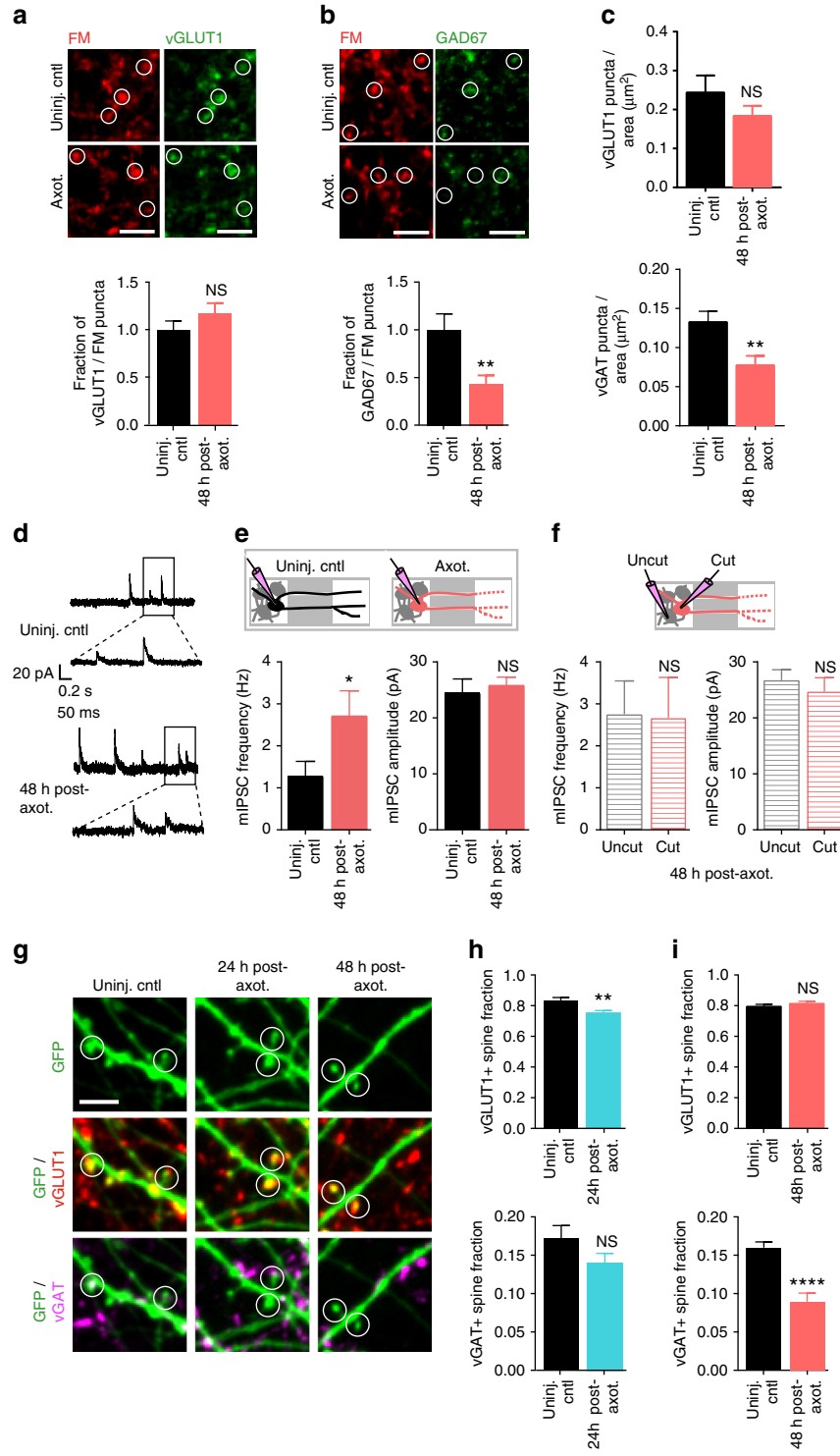

the axonal compartment in which axons were treated with a local activity blockade during axotomy. This local activity blockade solution (ABS) included low-$Ca^{2+}$, high-$Mg^{2+}$, and tetrodotoxin citrate (TTX; 0.5 mM $CaCl_2$, 10 mM $MgCl_2$, 1 μM TTX) to prevent influx of sodium and reduce calcium influx. This local activity blockade was applied solely to the axonal compartment for 1 h during axotomy. We labeled neurons extending axons into the axonal compartment using a retrograde eGFP rabies virus and quantified spine density before and 24 h following axotomy and compared these measurements to cultures with vehicle applied to axons during axotomy (Fig. 6a, b). Strikingly, we found that local activity blockade at the injury site prevented axotomy-induced spine loss. These data suggest that local activity instructs retrograde signaling and spine loss.

To determine whether injury-induced transcription is required for these trans-synaptic changes, we treated the somatodendritic compartment with the reversible transcriptional blocker 5,6-dichloro-1-β-D-ribofuranosyl-1H-benzimidazole (DRB) 15 min prior to axon injury and removed the drug 45 min later. We found that blocking transcription during this brief time was sufficient to prevent axotomy-induced spine loss 24 h post-axotomy compared with similarly treated uninjured control chambers (Fig. 6c). Further, DRB treatment at the time of injury prevented significant changes in the proportion of responsive FM puncta (Fig. 6d) and in synaptic vesicle release rate 48 h post-axotomy (Fig. 6e). However, action potential blockade with TTX in the somatodendritic compartment for ~1 h at the time of injury did not affect injury-induced changes in presynaptic release or the proportion of responsive puncta 48 h after axotomy (Fig. 6f, g). Further, application of Hank's balanced salt solution (HBS) or dimethyl sulfoxide(DMSO) as respective vehicle controls to TTX or DRB treatments did not alter injury-induced increase in presynaptic release. We conclude that both local activity at the site of injury and a transcriptional response were critical mediators of the delayed trans-synaptic changes in presynaptic release properties following distal axon injury.

**Differential gene expression at 24 h post-axotomy**. Our data show that a transcriptional response was required immediately after axotomy to induce retrograde changes in synaptic vesicle release onto injured neurons. To identify genes that might mediate this process within a longer therapeutically relevant time window, we performed a gene expression study to identify differentially expressed transcripts within the somatodendritic compartment at 24 h post-axotomy compared to uninjured controls (Supplementary Fig. 5). We found 615 transcripts that

were significantly changed following injury (one-way between-subject ANOVA, $p < 0.05$) (Fig. 7a; Supplementary Table 2). Confirming that the transcription response in vitro recapitulated in vivo findings, we found Jun upregulated 1.41 fold in our microfluidic cultures 24 h post-axotomy[19].

**Netrin-1 mRNA downregulated post-axotomy**. Next we sought to identify potential trans-synaptic mediators that may influence synaptic vesicle release at synapses onto injured neurons. We focused on differentially expressed transcripts that are known to localize to cell–cell contacts, such as synapses (Supplementary Table 3). We identified netrin-1 (Ntn1) as significantly downregulated 24 h following axotomy, consistent with published findings that netrin family proteins are downregulated following injury in adult rats[34] (Fig. 7b). Netrin-1 is a secreted axon guidance and synaptogenic cue that is enriched at mature dendritic spines[35] where it induces clustering of its receptor, deleted in colorectal carcinoma (DCC), and enhances synapse maturation[36]. To confirm that Ntn1 expression is downregulated following nerve injury in vivo, we analyzed microarray data from a previously published study which examined cortical gene expression from retrograde labeled layer V cortex of young adult rats (2 months) subjected to either sham injury or spinal cord hemisection at thoracic level 8[37] (Fig. 7c). Our analysis of this raw data confirmed that spinal cord injury significantly reduced netrin-1 expression in cortical layer V by 7 days, consistent with our in vitro findings. Together, the significant decrease in Ntn1 expression both in vitro and in vivo suggests a reliable response induced by distal axonal damage.

**Exogenous netrin-1 normalizes injury-induced changes**. The downregulation of netrin-1 following injury led us to ask whether adding exogenous netrin-1 might rescue, to some degree, the axotomy-induced synaptic changes. To test this we applied exogenous netrin-1 to the somatodendritic compartment 40 h after axotomy and evaluated the resulting changes in spine density, synaptic vesicle responsiveness, and disinhibition. We performed live imaging of somatodendritic domains before and after axotomy to measure spine density changes and found that netrin-1 treatment for 8 h was sufficient to normalize spine density to pre-axotomy levels (Fig. 8a, b). We then used FM dyes to compare presynaptic release properties between axotomized and uninjured controls. Exogenous netrin-1 increased the total number of FM puncta at 48 h post-injury to levels found in uninjured controls and reduced the percentage of responsive puncta to levels found in uninjured controls (Fig. 8c, d). We next tested whether netrin-1 might rescue axotomy-induced

**Fig. 5** Absence of inhibitory terminals following distal axotomy and frequency of their spontaneous release events. **a** Representative images of fixable FM4-64FX puncta (*red*) and vGLUT1 (*green*) co-immunolabeling in uninjured chambers and 48 h following axotomy. *White circles* highlight vGLUT1 expression at FM-labeled terminals. *Scale bars,* 10 μm. Fraction of vGLUT1 + FM puncta per neuron field at 48 h post-axotomy normalized to uninjured controls. $n = 18$ neuron fields; 5 chambers over 3 experiments. **b** Fixable FM4-64FX puncta (*red*) and GAD67 (*green*) co-immunolabeling. Quantification of GAD67-positive FM puncta at 48 h post-axotomy normalized to control. $n = 21$ neuron fields; 5 chambers over 3 experiments. **c** Number of vGLUT1 and vGAT puncta per neuron area (axotomized or control) 48 h post-axotomy. $n = 8$–9 neurons; 3 chambers per condition over 3 experiments. **a**–**c** Unpaired two-tailed t-test. **d** Representative traces of mIPSC recordings 48 h post-axotomy. **e** Quantification of mIPSC frequency and amplitude 48 h post-axotomy (control, $n = 9$ neurons; axotomy, $n = 17$ neurons). (mIPSC frequency: unpaired two-tailed *t*-test with Welch's correction, $p = 0.05$; mIPSC amplitude: unpaired two-tailed *t*-test, $p = 0.62$). **f** Analysis of mIPSC frequency and amplitude from axotomized devices previously shown in **e** comparing neurons with axons that extended into the axonal compartment and were cut (cut, $n = 8$ neurons), to neurons that did not extend axons into the compartment and were not cut (uncut, $n = 9$ neurons). (mIPSC frequency: unpaired two-tailed *t*-test, $p = 0.94$; mIPSC amplitude: unpaired two-tailed *t*-test, $p = 0.51$). **d**–**f** Data shown were combined from 3 chambers. **g** Representative dendritic segments (retrogradely labeled with eGFP rabies virus) showing spines that are labeled with vGLUT1 (*red*) or vGAT (*magenta*) antibodies. *White open circles* highlight dendritic spines with vGLUT1 and/or vGAT synapses. **h** Fraction of vGLUT1- and vGAT-positive spines at 24 h post-axotomy. $n = 12$–14 neuron fields; 7 neurons per condition; #spines: 458 (uninj. cntl), 394 (axot.); 3 chambers over 3 experiments. **i** Fraction of vGLUT1- and vGAT- positive spines at 48 h post-axotomy. $n = 11$–15 neuron fields; 8–9 neurons per condition; #spines: 460 (uninj. cntl), 413 (axot.); 3 chambers over 3 experiments. *Scale bars,* 5 μm. Unpaired two-tailed *t*-test. \*\**p* < 0.01, \*\*\**p* < 0.001. *Error bars,* s.e.m

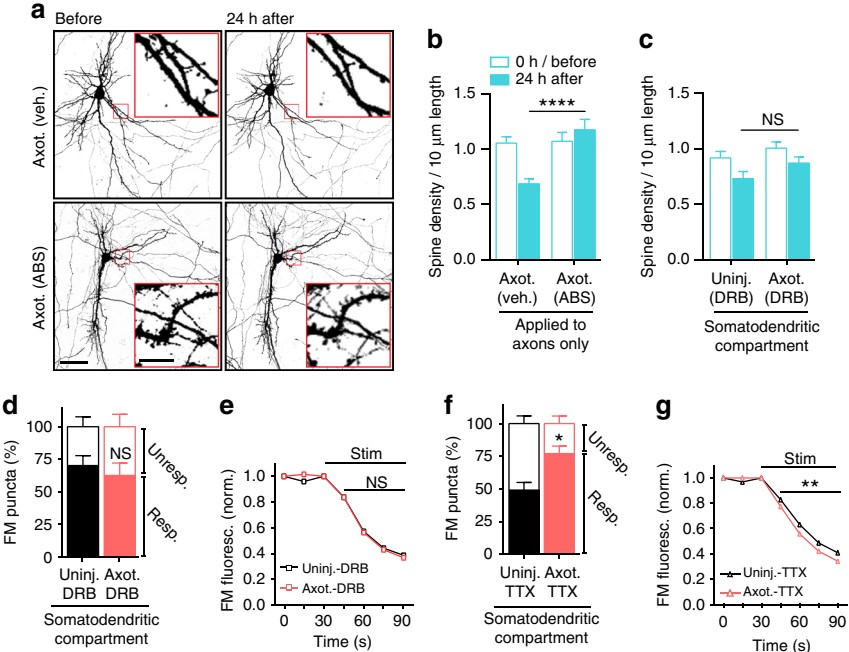

**Fig. 6** Influence of local activity and transcription on the initiation of injury-induced synaptic remodeling. **a** Representative images of neurons within microfluidic chambers retrograde labeled with eGFP rabies virus (*inverted grayscale*) before and 24 h post-axotomy with vehicle or local activity blockade solution (ABS) applied to axons for 1 h during axotomy. Inset shows zoomed in dendritic regions. Image and inset *scale bars*, 50 and 5 μm, respectively. **b** Quantification of before and after spine density data described in **a**. Axotomy (vehicle): n = 29 dendrites; 5 neurons; 3 chambers over 3 experiments; #spines/TDL: 437/4185 μm (before), 274/3868 μm (after). Axot. (ABS): n = 33 dendrites; 5 neurons; 3 chambers over 3 experiments; #spines/TDL: 426/3889 μm (before), 446/3700 μm (after). **c** Quantification of spine density changes following application of transcription blocker (DRB) to the somatodendritic compartment for 1 h during axotomy within microfluidic chambers. DRB (uninj.): n = 27 dendrites; 6 neurons; 3 chambers over 3 experiments; #spines/TDL: 362/4186 μm (0 h), 262/3934 μm (24 h after). DRB (axot.): n = 28; 6 neurons; 3 chambers over 3 experiments; #spines/TDL: 343/3414 μm (before) 283/3248 μm (after). **b**, **c** Unpaired two-tailed *t*-test. **d** Percentage of responsive and unresponsive FM puncta at 48 h post-axotomy. n = 6 neuron fields/chambers per condition over 3 experiments. Unpaired two-tailed *t*-test for % responsive. **e** FM 5–95 unloading following 1 h application of DRB during axotomy/control. Uninj. DRB: n = 1580 puncta; 6 chambers per condition over 3 experiments. Axot. DRB: 2213 puncta; 6 chambers per condition over 3 experiments. **f** Percent of responsive and unresponsive puncta at 48 h post-axotomy following application of TTX to the somatodendritic compartment for 1 h during injury. n = 6 neuron fields/chambers per condition over 3 experiments. Unpaired two-tailed t-test, % responsive. **g** FM unloading curves following application of TTX. Uninj. TTX: n = 1360 puncta; 6 chambers per condition over 3 experiments. Axot. TTX: n = 1648 puncta; 6 chambers per condition over 3 experiments. Two-way ANOVA, Bonferroni post hoc test. *p < 0.05, **p < 0.01, ***p < 0.001, ****p < 0.0001. *Error bars*, s.e.m

disinhibition. Netrin-1 treatment following axotomy normalized the density of inhibitory terminals (vGAT labeled) at axotomized neuron without significantly altering the density of glutamatergic terminals (vGLUT1 labeled) (Fig. 8e, f).

Because DCC protein levels parallel netrin-1 expression changes[38, 39], we next confirmed that DCC levels were downregulated at synapses formed onto the somatodendritic domain of axotomized neurons (Fig. 8g, h). Local synaptic DCC immunofluorescence at spines of axotomized neurons were decreased at 48 h post-injury. Further, application of exogenous netrin-1 normalized synaptic DCC levels to that similar to uninjured controls (Fig. 8g, h).

If downregulation of netrin-1 signaling regulates axotomy-induced synaptic remodeling, we would expect that blocking netrin-1 signaling in uninjured neurons would be sufficient to cause both reductions in spine density and the density of inhibitory terminals. Spine density in uninjured cultures treated with a DCC function blocking antibody for 24 h was significantly reduced after treatment compared to control chambers treated with an IgG antibody (Fig. 8i). Further, we found that blocking DCC was sufficient to cause a reduction in the density of vGAT puncta, but not vGLUT1 puncta, per eGFP-filled neuron area (Fig. 8j). Together, our data suggest that netrin-1 signaling may play a critical role in regulating synaptic remodeling

following axonal damage, including in modulating inhibition following injury.

## Discussion

While axon regeneration following injury is extensively studied, much less is known about how proximal neurons within the mammalian brain are affected following axonal damage[40] and more specifically how synapses onto injured neurons are remodeled. We used a model system to study the cellular mechanisms of synaptic remodeling following axon injury; this model recapitulated several hallmarks of neurons subjected to axonal injury in vivo, including chromatolysis[6, 21], retrograde spine loss[4, 22, 23], retrograde hyper-excitability[1–3], and disinhibition[2, 9, 10]. Axotomy-induced transcriptional changes in this in vitro model are also consistent with in vivo findings[7, 20]. Because of the ability to separate neuronal compartments, this tool facilitates the investigation of axotomy-induced retrograde signaling intrinsic to neurons and the resulting effects to interneuronal communication.

Our results suggest that retrograde remodeling requires local signaling at the site of injury mediated by sodium and/or calcium influx to activate a rapid transcriptional response. Both postsynaptic dendritic spine loss and trans-synaptic changes in

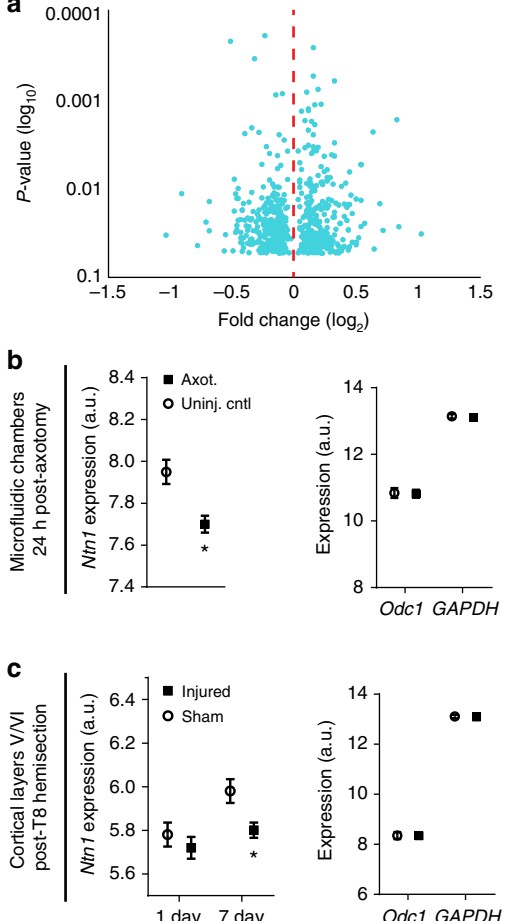

**Fig. 7** Netrin-1 gene expression following axotomy in microfluidic cultures and in vivo following spinal cord hemi-transection Microarray analysis was performed on somatodendritic samples of uninjured control and 24 h post-axotomy chambers. Quality control data is presented in Supplementary Fig. 5. **a** Volcano plot showing differentially expressed RNAs that are significantly changed at 24 h post-axotomy (One-way between-subject ANOVA; $n = 3$ individual chambers each condition; Supplementary Table 2). **b** Microarray expression levels for *Ntn1* within microfluidic chambers (*left*) and for housekeeping genes *Odc1* and *GAPDH* (*right*). One-way between-subject ANOVA. **c** Microarray expression levels for *Ntn1* (*left*) and *Odc1* and *GAPDH* (*right*) in cortical layers V/VI following hemi-transection at thoracic level 8. $n = 4$ animals per group. *Ntn1* levels are significantly reduced by 7 days following injury. Two-way ANOVA, Sidak's multiple comparisons test. *$p < 0.05$. Error bars, s.e.m

presynaptic inputs required immediate transcription. Our data is consistent with axonal injury signaling in other non-CNS model systems[6]. Localized reversal of a sodium calcium exchanger at the site of injury may amplify calcium influx and contribute to long range signaling[41]. In peripheral neurons calcium waves can locally propagate to the nucleus to induce a transcriptional response[42]. The localized influx of calcium may be a priming effect for retrograde transport of signaling complexes required to initiate transcription[6].

Our data showed that axotomy-induced spine loss was followed by a specific loss of inhibitory inputs. Interestingly, specific loss of inhibitory, and not excitatory, terminals suggests that preserved excitatory inputs may remain available for some period of time following injury. Because of the spine loss, these excitatory inputs could form shaft synapses or some may become

orphan presynaptic sites following injury. Large headed dendritic spines could also receive multiple excitatory inputs, stabilizing them, and allowing them to find new partners over time. The increased spontaneous release of glutamate at injured neurons 48 h following axotomy, without an increase in the number of excitatory terminals, suggests that the maintained excitatory inputs may contribute to the hyper-excitability post-injury.

The sequential post- and then pre- synaptic changes following axotomy suggest a trans-synaptic mechanism. These post- and then pre-synaptic changes are consistent with the involvement of synaptic homeostasis where retrograde molecules are released post-synaptically to influence presynaptic release. Further support for the involvement of trans-synaptic mechanisms comes from our observation of an increase in spontaneous neurotransmitter release localized at excitatory synapses onto axotomized neurons, but not at neighboring excitatory synapses onto uninjured neurons. The significantly enhanced firing rate following SCI in cortical layer Vb, but not layers Va and VI, provides additional support for this specificity. Interestingly, we also found an increase in spontaneous release at inhibitory terminals, although fewer inhibitory terminals remain following axotomy. The increase in GABA release rate may serve to compensate for the axotomy-induced hyper-excitability.

Dendritic release of secreted proteins (e.g., BDNF, NT-3, and NT-4) and diffusible molecules, such as nitric oxide, can trans-synaptically regulate neurotransmitter release[43–45]. Injury of motoneuron projections to myocytes caused synaptic remodeling of inputs to motoneurons which was influenced by nitric oxide synthesis[46]. While these previously reported trans-synaptic signaling pathways were not detectably altered in our micro-array analysis, we did identify the secreted protein, netrin-1, as significantly downregulated in our axotomized cultures 24 h post- axotomy. Downregulation of netrin-1 gene expression was further confirmed in vivo through an analysis of independently acquired microarray data. Netrin-1 is secreted locally from target cells and signals DCC receptors that are present along axons[36] to influence presynaptic release and maturation[47, 48]. While netrin-1 signaling is historically thought of in a developmental context, there is increasing evidence of the importance of netrin-1 signaling in the adult CNS. Consistent with our in vitro findings, netrin family members are downregulated in vivo following spinal cord injury in adult rats[34, 49] and DCC remains persistently low after 7 months post-injury in adult rats[34]. Netrin-1 has also recently been tested as a potential therapeutic agent following injury and has been shown to improve recovery outcomes[50–52].

We found that adding exogenous netrin-1 one and a half days after axotomy dramatically increased spine density and the density of inhibitory terminals to levels found in uninjured controls. The restoration of inhibitory terminals in axotomized samples treated with netrin-1 is a novel and exciting finding. Evidence from *C.elegans* confirms a link between netrin-1 signaling and stabilization of GABA$_A$ receptors[53]. Yet, it remains unclear how netrin-1 signaling modulates inhibitory input and will be an important topic for future studies. In contrast to previous reports, we found that there was no reduction in the number of vGLUT1 positive terminals when blocking DCC[36], which could be explained by our shorter treatment times with DCC function blocking antibody.

Axonal damage within the CNS occurs in numerous disorders and diseases, but little is known about the overall impact on cortical circuit function. Importantly, our cell-based findings have broader applicability beyond spinal cord injury to numerous conditions where axonal damage is prevalent, such as other forms of traumatic brain injury, Alzheimer's

disease, and multiple sclerosis. Further, remodeling is enhanced in embryonic or neonatal neurons, making the use of an in vitro approach using these neurons, together with in vivo models, advantageous for identifying pathways instrumental for neurological recovery[54].

## Methods

**Hippocampal cultures**. Animal procedures were approved by the University of North Carolina at Chapel hill Institutional Animal Care and Use Committee (IACUC). Dissociated hippocampal cultures were prepared from Sprague Dawley rat embryos (E18-E19)[11, 24]. Hippocampal tissue was dissected in ice cold dissociation media (DM) containing 82 mM $Na_2SO_4$, 30 mM $K_2SO_4$, 5.8 mM

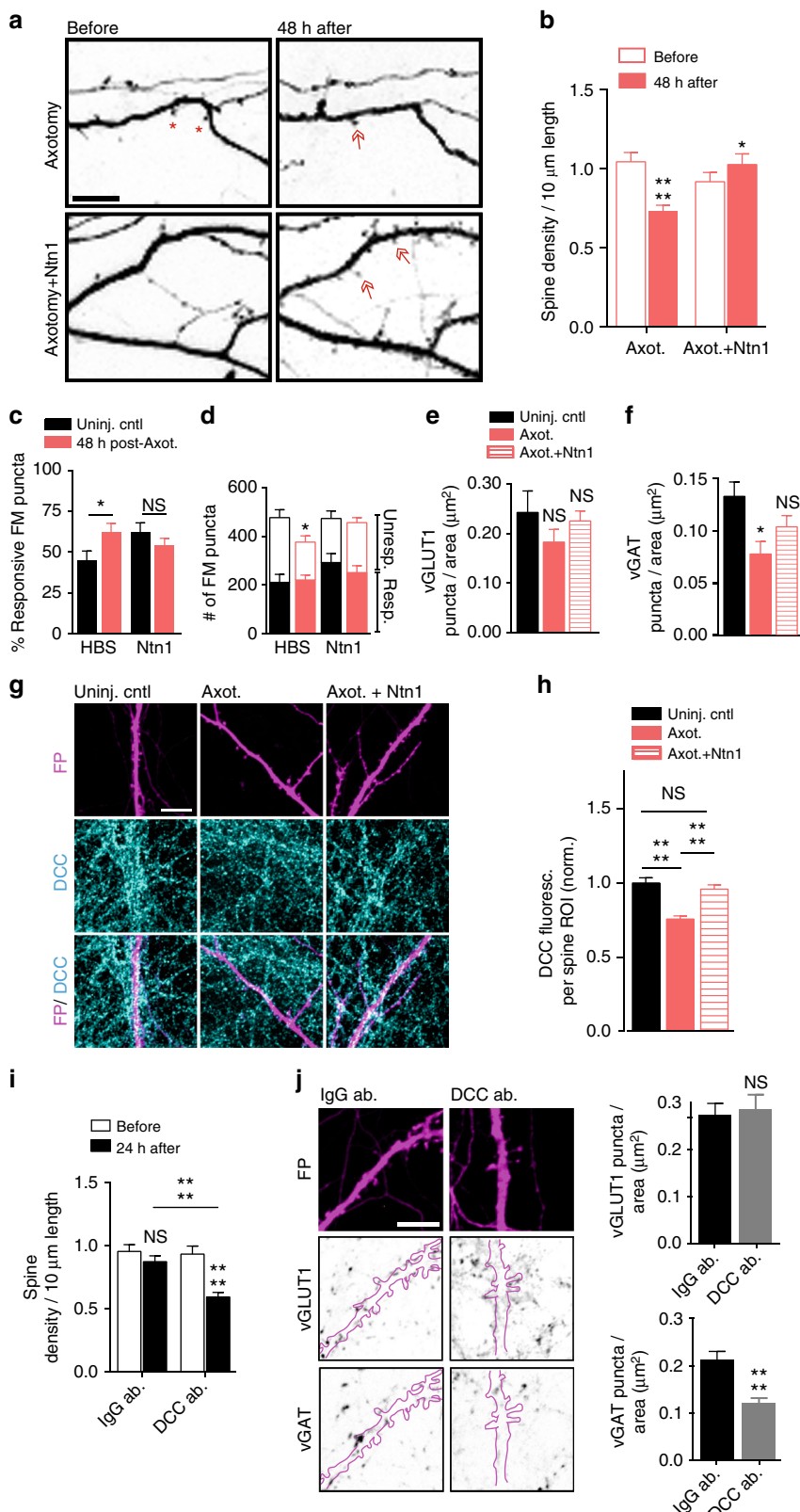

MgCl$_2$, 0.25 mM CaCl$_2$, 1 mM HEPES, 20 mM Glucose and 0.001% Phenol red. For enzymatic digestion, equal volumes of TrypLE Express (Invitrogen) and DM were added to the tissue and incubated at 37 °C for 8 min. Tissue was then rinsed and gently triturated in neuronal culture media consisting of Neurobasal media (Invitrogen) supplemented with 1 × B27 (Invitrogen), 1 × Antibiotic-antimycotic (Invitrogen), and 1 × Glutamax (Invitrogen). Dissociated cells were resuspended in neuronal culture media to yield 12 × 10$^6$ cells per ml.

**Microfluidic chambers.** Poly(dimethylsiloxane) (PDMS) (Sylgard 184 Silicon Elastomer, Dow Corning) microfluidic chambers were replica molded against silicon wafers photolithographically patterned with SU-8 negative photoresist (MicroChem)[11]. Each silicon wafer contained a first layer of SU-8 to pattern microgrooves 3–4 µm tall and 7.5–8 µm wide. A second layer of SU-8 generated the 85–100 µm high somatodendritic and axonal compartments. All experiments used chambers with 900 µm long microgrooves to separate the somatodendritic and axonal compartments[11, 16, 24]. Microfluidic chambers were sterilized using 70% ethanol and placed onto sterile German glass coverslips coated with 500–550 kDa Poly-D-Lysine (BD Biosciences). Approximately ~90,000 cells were plated into the somatodendritic compartment and axons extended into the adjacent axonal compartment after 5–7 days of culture. Axotomy was performed between 11 and 15 days in vitro (DIV) by first removing media from the axonal compartment and storing for future use. The axonal compartment was then aspirated until completely devoid of fluid[11, 16]. Stored culture media was returned immediately to the axonal compartment for the duration of the culture time. Microfluidic devices with equivalent viable cell populations were randomly chosen for either axotomy or uninjured control groups.

**Retrograde labeling.** Retrograde labeling was performed using either modified cholera toxin or rabies virus. Cholera Toxin Subunit B Alexa Fluor 488 or 568 (Life technologies, Molecular Probes; 1 µg in 200 µl of neuronal culture media) was added to the axonal compartment of the microfluidic chamber and incubated for ~15 h at 37 °C. After 15 h of incubation, the axonal compartment media was removed, rinsed and replaced using fresh neuronal culture media before performing axotomy or imaging.

G-deleted Rabies-mCherry or eGFP virus[55] (Salk Institute; 1 × 10$^5$ viral units) in 50 µl- conditioned media was added to the axonal compartment of each chamber and incubated for 2 h at 37 °C. Conditioned media was added back to the axonal compartments following two washes with fresh Neurobasal media. Chambers were maintained in 37 °C incubator for ~48 h until fluorescence expression was visible.

**Cell viability assay.** Dead cells were labeled using SYTOX Green (Invitrogen) at a final concentration of 1 µM and all cell nuclei were labeled with NucBlue Hoechst Stain (Invitrogen). Cells were incubated with SYTOX/Hoechst solution simultaneously in 1xPBS for 5 min at 37 °C, washed with PBS, and fixed with 4% paraformaldehyde (PFA) in PBS containing 40 mg/ml sucrose, 1 µM MgCl$_2$, and 0.1 µM CaCl$_2$ for 15 min at room temperature (RT). Coverslips were then rinsed three times with PBS and mounted onto the glass slide using Fluoromount G (Southern Biotech). SYTOX positive (Sytox$^+$) cells were manually counted in ImageJ using sum projected z-stack confocal images. Percent cell viability is calculated using [(Hoechst - Sytox$^+$)/Hoechst]×100.

**Nissl Staining.** Neuronal cultures retrogradely labeled with Cholera Toxin were either axotomized or left uninjured. PDMS chambers were carefully lifted off from

PDL coated coverslips 24 h post-axotomy. Cultures on the coverslips were quickly rinsed twice with PBS, fixed with 4% PFA for 30 min at RT, washed twice in PBS, and incubated in 0.1% Triton X-100/PBS for 10 min at RT. Cultures were incubated for 20 min in NeuroTrace 500/525 Green Fluorescent Nissl Stain (1:100; Invitrogen) and washed for 10 min in 0.1% Triton X-100/PBS. Cell nuclei were stained with DAPI (Sigma-Aldrich), rinsed three times in PBS, and then the coverslip was mounted onto a microscope slide using Fluoromount G.

**Immunocytochemistry.** PFA fixed neuronal cultures were permeabilized in 0.25% Triton X-100 and blocked in 10% normal goat serum for 15 min each. Coverslips were incubated with anti-MAP2 (1:1000; Millipore # AB5622), anti-beta tubulin III (1:2000; Aves #TUJ), anti-GAD67 (1:2000; Aves labs # GAD), anti-vGLUT1 (1:100; NeuroMab, clone N28/9, cat. #75-066), anti-vGAT (1:1000; Synaptic Systems #131 003), anti-DCC (1:100; Calbiochem #OP45), or anti-synapsin1 (1:500; Calbiochem #574778) primary antibodies in 1% blocking solution for overnight at 4 °C. Coverslips were then incubated with goat anti-rabbit or goat anti-mouse or anti-chicken secondary antibodies conjugated to Alexa-fluorophores (1:1000; Invitrogen) for 1 h at RT. Following PBS washes coverslips were mounted onto glass slides.

**RNA isolation.** Total RNA from each of 3 axotomized chambers and 3 sham manipulated chambers (6 total samples) was isolated from the somatodendritic compartment of 14 DIV cultures, 24 h after manipulation. RNA was collected from the entire somatodendritic compartment for our gene expression analysis; thus, a fraction of neurons in the axotomized chambers were axotomized and the remaining fraction uninjured or "uncut". RNA was isolated using an RNAqueous-Micro Kit (Ambion) according to the manufactures instructions including DNase treatment, with modifications specific to accessing the microfluidic compartment[16]. Briefly, 50 µl lysis solution was added to one somatodendritic well and collected from the other somatodendritic well after solution flowed through the somatodendritic compartment to this adjacent well. Lysate was added to 50 µl of fresh lysis solution and mixed well by careful pipetting. Further RNA purification steps were performed according to the manufacturer's guidelines. Samples were maintained at −80 °C until prepared for microarray gene expression.

**Microarray analysis.** Quantification of RNA integrity and concentration was confirmed with an Agilent TapeStation 2200 at the UNC Lineberger Comprehensive Cancer Center Genomics Core. Microarrays were processed at the UNC School of Medicine Functional Genomics Core using the Affymetrix GeneChip WT Plus Reagent Kit for cRNA amplification, cDNA synthesis, fragmenting and labeling. Samples were hybridized to Rat Gene 2.0 ST Arrays (Affymetrix). Data analysis was performed with Affymetrix Expression Console software and Affymetrix Transcriptome Analysis Console v2.0 software to compare axotomized cultures to uninjured control samples using one-way between-subject ANOVA of Robust Multi-array Average (RMA) normalized intensities. Quality control data is presented in Supplementary Fig. 5. Because a fraction of the harvested cells were uninjured in our axotomized samples, we used modest fold change values for defining our list of significantly changed transcripts (fold change absolute value ≥ 1.1 and ANOVA $p$-value < 0.05). To identify cell–cell adhesion transcripts we searched the biological process gene ontology category "cell–cell adhesion". Fold change shown in Fig. 7 was calculated by dividing the mean log$_2$ intensity value of the uninjured control by the mean log$_2$ intensity value of the axotomized culture samples.

---

**Fig. 8** Spine density and presynaptic release properties of axotomized neurons treated with exogenous netrin-1. **a** Representative dendrites before and 48 h post-axotomy treated with vehicle (HBS) or netrin-1 (Ntn1) beginning at 40 h post-axotomy (inverted fluorescence). *Arrows*: new spines; *red asterisks*: eliminated spines. *Scale bars*, 10 µm. **b** Quantification of spine density illustrated in **a**. *Axotomy*: n = 33 dendrites; 7 neurons; #spines/TDL: 392/3696 µm (*before*), 263/3447 µm (*after*). *Axotomy + netrin-1*: n = 29 dendrites, 6 neurons; #spines/TDL: 293/3281 µm (*before*), 363/3417 µm (*after*). **c** Percent responsive FM puncta per neuron field at 48 h post-axotomy with HBS or netrin-1. n = 8–11 fields/chambers per condition over 5 experiments. **d** Number of responsive and unresponsive FM puncta from **c**. Significantly fewer unresponsive puncta followed axotomy compared to uninjured control (HBS). **e**, **f** Number of vGLUT1 and vGAT puncta per neuron area (uninjured control, axotomized + HBS, or axotomized + netrin-1) at 14 DIV. n = 8–9 neurons; 3 chambers per condition over 3 experiments. **g** Representative DCC immunostaining (*turquoise*) in uninjured control, post-axotomy, and post-axotomy + netrin-1 in cultures with similar spine densities. Neurons were retrogradely labeled with fluorescent protein (FP, *magenta*) using an mCherry modified rabies virus. *Scale bar*, 10 µm. **h** Quantification of DCC immunofluorescence per spine region-of-interest (ROI). ROI: 2 µm diameter circular region surrounding each spine. Control, n = 295 ROIs; axotomy, n = 293 ROIs; axotomy + Ntn1, n = 210 ROIs. 8 neuron fields/3 chambers per condition; 3 experiments. **i** Quantification of spine density following 24 h of control antibody (IgG ab.) or DCC function blocking antibody (DCC ab.). IgG: n = 33 dendrites; 8 neurons; #spines/TDL: 464/4864 µm (before), 419/4712 µm (after). DCC ab: n = 34 dendrites; 7 neurons; #spines/TDL: 404/4433 µm (before), 222/3647 µm (after). **j** Representative FP-labeled dendritic segments immunostained for vGAT (inverted) and vGLUT1 (inverted) following 24 h application of IgG or DCC antibodies (outlined dendrites, *solid magenta line*). Neurons were fixed at 15–16 DIV, older than the cultures in **e**, **f**. *Scale bar*, 10 µm. Quantification shown on the right. n = 23 neuron fields per condition; 3 chambers per condition over 3 experiments. **b**, **i** Repeated-measure two-way ANOVA, Bonferroni post hoc test; analyses included 1 chamber per condition for 3 experiments. **c**, **j** Unpaired two-tailed $t$-test. **d**–**f**, **h** One-way ANOVA, Bonferroni post hoc test. *Error bars*, s.e.m. *$p$ < 0.05, ****$p$ < 0.0001

Raw microarray data of cortical layers V/VI of female Wistar rats subjected to either spinal cord transections at thoracic layer 8 or sham injury 1 day and 7 days following injury was downloaded from EMBL-EBI Array Express (E-MTAB-794)[37]. Four animals were used for each condition and samples were hybridized to Rat Gene 1.0 ST Arrays (Affymerix). Data analysis was performed with Affymetrix Expression Console software and Affymetrix Transcriptome Analysis Console v2.0 software to compare cortical layers V/VI from lesioned to sham operated animals.

**Image acquisition and dendritic spine analysis**. High-resolution z-stack montages of mCherry or eGFP labeled live neurons were captured using either a Zeiss LSM 780 (63 × 1.4 NA or 40 × 1.4 NA oil immersion objective) or an Olympus IX81 microscope (60 × 1.3 NA silicon oil immersion objective). For live imaging, we captured "0 h" or "before axotomy" confocal z-stack images to create montages of neurons extending axons into the axonal compartment. Axotomy was performed on the same day after acquiring these images. Images were acquired from the same neuron 24 h post-axotomy. In some cases, images were also acquired from the same neurons at 48 h post-axotomy (Figs. 1, 2, and 8). Calibrated z-stack montages were analyzed for all dendrite and spine parameters. Primary dendrites were traced using the semiautomatic neurite tracing tool, Neuron J[56, 57]. The number of spines on all primary dendrites of each neuron were manually labeled and categorized as thin, stubby or mushroom shaped using Neuron Studio[58]. Spine density was calculated for 10 μm length of dendrite as [(# of spines/dendrite length)×10]. Blinded data analysis was perfomed.

**FM dye experiments and analysis**. Cultures in microfluidic chambers at 24 h (14 DIV), 48 h (15 DIV), and 4 d (17 DIV) post-axotomy were loaded with lipophilic dye N-(3-trimethylammoniumpropyl) -4-(6-(4-(diethylamino) phenyl) hexatrienyl)pyridinium dibromide (FM 5–95; Invitrogen) using KCl mediated depolarization[24]. Cultures were first incubated for 30 min with pre-warmed HEPES-buffered solution (HBS; 119 mM NaCl, 5 mM KCl, 2 mM CaCl₂, 2 mM MgCl₂, 30 mM glucose, and 10 mM HEPES). Media was then replaced with FM dye loading solution containing 10 μM FM 5–95, 20 μM AMPAR antagonist 6-cyano-7-nitroquinoxaline-2,3-dione disodium (CNQX; Tocris), 50 μM NMDAR antagonist D-(-)-2-amino-5-phosphonopentanoic acid (D-AP5; Tocris) in 90 mM KCl HBS for 1 min. The loading solution was replaced with HBS containing 10 μM FM 5–95 for 1 min and later rinsed three times with a high-Mg²⁺, low-Ca²⁺ solution (106 mM NaCl, 5 mM KCl, 0.5 mM CaCl₂, 10 mM MgCl₂, 30 mM glucose, and 10 mM HEPES) containing 1 mM Advasep-7 (Biotium) to remove extracellular membrane-bound FM. Finally, cultures were washed in HBS containing 20 μM CNQX and 50 μM D-AP5 for at least three times, 1 min each. Next, we stimulated the microfluidic chambers using extracellular electrodes by placing a positive and negative electrode in each well of the somatodendritic compartment.

Electrical stimulation was provided by an AD Instrument 2 Channel Stimulus Generator (STG4002) in current mode with an asymmetric waveform (−480 μA for 1 ms and + 1600 μA for 0.3 ms) for ~ 1 min at 20 hz for 600 pulses. The FM 5–95 imaging was performed using a spinning disk confocal imaging system[24]. Z-stacks (31 slices) were captured every 15 s during the baseline (1 min), stimulation (1 min), and after stimulation (2 min) periods. This stimulation pattern was optimized for efficient FM unloading within these microfluidic chambers and the frequency is greater than typically used in open well dishes. At least 3 baseline images were acquired before electrical stimulation.

Blinded data analysis was perfomed. Sum projected confocal z-stack were converted to 8-bit images and registered using TurboReg, an Image J plugin. We background subtracted the image stack using the image 3 min after stimulation began. Image stacks were thresholded to a pixel value of 15. FM puncta between 0.4 to 10 μm² were analyzed. We measured the intensity of each punctum in the whole field or specifically on eGFP labeled neurons (Fig. 3a–c) throughout all time series. To analyze the unloading kinetics of FM puncta on eGFP labeled neurons, we first thresholded the eGFP image and then created an outline enclosing all the eGFP labeled regions including spines. The outlined ROI was superimposed on the FM-labeled image and the intensity of each punctum in the selected ROI (eGFP outline) was measured throughout all time series. We normalized fluorescence intensity of each puncta to the frame before stimulation. Puncta with >5% unloading after 1 min were used in the analysis as unloaded puncta. Time constants were estimated by curve fitting unloading kinetics to a single exponential decay function[24]. Curve fitting was done in MATLAB and FM puncta with time constants longer than 3 min were excluded from the analysis and assumed to be non-releasing. Number of FM puncta that unload >5% after 60 s were classified as responsive using image stacks that were not background subtracted; puncta that did not meet this criteria were classified as unresponsive.

In activity and transcription blocking experiments, the FM 5–95 unloading experiment was performed as mentioned above at 48 h post-axotomy. The intensity measurements of each punctum in the whole field and subsequent analysis of FM unloading kinetics was performed as mentioned above.

To determine the fraction of vGLUT1 or GAD67-positive FM puncta, the somatodendritic compartment of 15 DIV cultures were loaded with FM 4-64FX (10 μM; Invitrogen), fixable analog of N-(3-triethylammoniumpropyl) -4-(6-(4-(diethylamino) phenyl)hexatrienyl)pyridinium dibromide (FM 4–64) membrane stain, using KCl mediated depolarization as described above. Following subsequent strip and wash steps, cells were fixed with 4% PFA in PBS and immunostained with

anti-GAD67 and anti-vGLUT1 antibodies. Total number of vGLUT1, GAD67, and FM puncta were acquired by processing confocal Z-stack images using 3D foci picker, an ImageJ plugin[59].

**Drug treatments**. Local ABS, which includes low-Ca²⁺, high-Mg²⁺, and TTX (0.5 mM CaCl₂, 10 mM MgCl₂, 1 μM TTX) was applied soley to the axonal compartment for 1 h during axotomy (15 min prior and 45 min after axotomy). DRB (Sigma-Aldrich # D1916) was suspended in DMSO and applied to the somatodendritic compartment at a final concentration of 80 μM for 1 h during axotomy (beginning 15 min prior to axotomy). Tetrodotoxin citrate (TTX; Tocris Bioscience #1078) was suspended in HBS and applied to the somatodendritic compartment at a final concentration of 1 μM for 1 h during axotomy (beginning 15 min prior to axotomy). Media stored from the axonal compartment prior to treatment was added back to the axonal compartment after treatment. Exogenous netrin-1 was applied to the somatodendritic compartment 1¹/₂ days after axotomy, when spine changes were observed, at a final concentration of 625 ng/ml. A similar netrin-1 concentration has been used for cortical neurons over a treatment time of 1–2 days to examine netrin-specific responses[60]. Netrin-1 was applied for 8–10 h to observe stable synaptic changes. For DCC function blocking experiments, anti-DCC (mDCC; Calbiochem # OP45), and isotype control (mIgG; BD pharmingen #554121) was applied to the somatodendritic compartment of uninjured chambers at a final concentration of 1 μg/ml for 24 h.

**Microscopy**. FM and fixed imaging was performed using CSU-X1 (Yokogawa) spinning disk confocal imaging unit configured for an Olympus IX81 microscope (Andor Revolution XD). Live imaging of neurons for spine analysis was captured using a Zeiss LSM 780 confocal microscope with a Plan-Apochromat 40 × objective (NA 1.4) at the UNC Neuroscience microscopy core facility. Excitation for the spinning disk confocal imaging system was provided by 405 nm, 488 nm, 561 nm, and/or 640 nm lasers. The following bandpass emission filters (BrightLine, Semrock) were used for the spinning disk: 447/60 nm (TRF447-060), 525/30 nm (TRF525-030), 607/36 nm (TR-F607-036), and 685/40 nm (TR-F685-040). For FM imaging, the spinning disk confocal imaging system was used with excitation at 561 nm and the 685/40 nm emission filter. We used 2 × 2 binning to reduce the laser intensity and acquisition time for each frame; each z-stack was obtained in ~5 s. For the Zeiss LSM 780, signal was acquired from eGFP (493 nm - 558 nm), Alexa 568 (569 nm–630 nm), and Alexa 647 (640 nm–746 nm).

**Whole-cell electrophysiology**. For whole-cell recordings, neurons were visually identified with infrared differential interference contrast optics. Cells were recorded in voltage-clamp configuration with a patch clamp amplifier (Multiclamp 700 A), and data were acquired and analyzed using pCLAMP 10 software (Molecular Devices). Patch pipettes were pulled from thick-walled borosilicate glass with open tip resistances of 2–7 MΩ. Series and input resistances were monitored throughout the experiments by measuring the response to a −5-mV step at the beginning of each sweep. Series resistance was calculated using the capacitive transient at the onset of the step and input resistance was calculated from the steady-state current during the step. Recordings were sampled at 10 kHz and bessel filtered at 2 kHz. No series resistance compensation was applied.

Prior to recording, microfluidic chambers and PDMS molds were removed and the glass coverslips containing cells were mounted onto a submersion chamber, maintained at 32 °C. Cultures were perfused at 2 ml/min with artificial cerebrospinal fluid (ACSF) containing 124 mM NaCl, 3 mM KCl, 1.25 mM Na₂PO₄, 26 mM NaHCO₃, 1 mM MgCl₂, 2 mM CaCl₂, and 20 mM d-( + )-glucose, saturated with 95% O₂ and 5% CO₂. To determine if recorded neurons' axons entered the microfluidic chamber, 0.035 mM Alexa-594 was included in all internal solutions to allow for post hoc visualization of neuronal morphology.

Events with a rapid rise time and exponential decay were identified as mEPSCs or mIPSCs respectively using an automatic detection template in pCLAMP 10[61]. mEPSC events were post-hoc filtered to only include events with a peak amplitude ≥5 pA and a ≤ 3 ms 10–90% rise time. Mean mEPSC parameters were quantified from a 10 min recording period and mIPSC parameters were sampled from a 5 min recording period. Neurons were excluded from analysis if $R_{series}$ was >25 MΩ anytime during the recording.

**mEPSC recordings**. AMPAR-mediated mEPSCs were isolated by voltage-clamping neurons at −70 mV and by supplementing the ACSF with TTX citrate (1 μM, Abcam), the GABA (A) receptor antagonist picrotoxin (50 μM, Sigma-aldrich), and the NMDA receptor antagonist D, L-2-amino-5 phosphonopentanoic acid (100 μM, AP5, Abcam)[62]. The internal solution contained: 100 mM CsCH₃SO₃, 15 mM CsCl, 2.5 mM MgCl₂, 5 mM QX-314-Cl, 5 mM tetra-Cs-BAPTA, 10 mM HEPES, 4 mM Mg-ATP, 0.3 mM Na-GTP, and 0.5% (w/v) neurobiotin with pH adjusted to 7.25 with 1 M CsOH and osmolarity adjusted to ~295 mOsm with sucrose.

**mIPSC recordings**. mIPSCs were isolated by supplementing the ACSF with TTX citrate (1 μM), the NMDA receptor antagonist AP5 (100 μM), and the AMPA/Kainate receptor antagonist 6,7-dinitroquinoxaline-2,3-dione (20 μM, DNQX, dissolved in DMSO for a final concentration of 1% v/v DMSO in ACSF, Abcam).

For mIPSC recordings, the pipette solution contained a relatively lower chloride concentration, similar to intracellular chloride concentrations that are present in more mature neurons[63]. This pipette solution contained, 110 mM $CsCH_3SO_3$, 2.5 mM $MgCl_2$, 5 mM QX-314-Cl, 5 mM tetra-Cs-BAPTA, 10 mM HEPES, 4 mM Mg-ATP, 0.3 mM Na-GTP, and 0.5% (w/v) neurobiotin with pH adjusted to 7.28 with 1 M CsOH and a 300 mOsm osmolarity. Following break-in, neurons were first voltage-clamped at −70 mV for at least 3 min to allow dialysis with pipette solution, after which the voltage was gradually changed to 0 mV, where it was maintained for duration of the recording.

**SCI injury and in vivo electrophysiology**. Nineteen adult male, Fischer-344 inbred rats (Harlan Laboratories, Indianapolis, IN, USA) were selected for this study. A total of 14 rats received a contusion injury in the thoracic cord at level T9–T10, whereas 5 rats were randomly selected as uninjured controls. After a minimum of 4 weeks following SCI, intracortical microstimulation (ICMS) and single-unit recording techniques were used in the hindlimb motor area (HLA) to determine movements evoked by ICMS and spike rates. The protocol was approved by the University of Kansas Medical Center Institutional Animal Care and Use Committee.

Spinal cord surgeries were performed under ketamine hydrochloride (80 mg/kg)/xylazine (7 mg/kg) anesthesia and aseptic conditions. Each of the SCI rats underwent a T9–T10 laminectomy and contusion injury using an Infinite Horizon spinal cord impactor (Precision Systems and Instrumentation, LLC, Fairfax Station, VA, USA) with a 200 Kdyn impact. At the conclusion of surgery, 0.25% bupivacaine hydrochloride was applied locally to the incision site. Buprenex (0.01 mg/kg, SC) was injected immediately after surgery and 1 day later. On the first week after surgery, the rats received daily injections of 30,000U penicillin in 5 ml saline. Bladders were expressed twice daily until the animals recovered urinary reflexes.

Post-SCI surgical and neurophysiological procedures were conducted under aseptic conditions 4–18 weeks post-SCI. At the time of these procedures, ages ranged from 4.5 to 7.5 months. After an initial, stable anesthetic state was reached using isoflurane anesthesia, isoflurane was withdrawn and the first dose of ketamine hydrochloride (100 mg/kg)/xylazine (5 mg/kg) was administered. The rats were placed in a Kopf small-animal stereotaxic instrument and a craniectomy was performed over the motor cortex. The dura was incised and the opening filled with warm, medical grade, sterile silicone oil. Core temperature was maintained within normal physiological limits using a feedback-controlled heating pad during the entire procedure. A stable anesthetic level was assessed by monitoring the respiratory and heart rate, and reflexes to noxious stimuli.

In each rat, neuronal recordings were begun at ~3 h after initiation of the procedure. Neuronal action potentials (single-units or spikes) were recorded with a single-shank, 16-channel Michigan-style linear microelectrode probe (Neuronexus, Ann Arbor, MI). A total of 15 channels were active in each procedure. The tip of the probe was lowered to a depth of 1720 μm below the cortical surface, allowing accurate determination of depth for each recording site. Because no hindlimb responses were evoked using ICMS in SCI rats, the location of HLA in SCI rats was determined by the stereotaxic coordinates derived in normal rats in a previous study (centered at 2 mm posterior and 2.64 mm lateral to bregma). At each cortical location, electrical activity was collected and digitized for 5 min from each of the 15 active sites using neurophysiological recording and analysis equipment (Tucker Davis Technologies, Alachua, FL, USA). Neural spikes were discriminated using principle component analysis. Sample waveforms (1.3 msec in duration) were collected that passed 5.5 × SD below root mean square. After each experiment, the probe was cleaned with Opti-Free solution (Alcon Laboratories, Fort Worth, TX, USA), followed by ethanol and then rinsed thoroughly in distilled water. The electrode impedance of each site remained at 0.9 MΩ for each experiment. At the end of the recording session, rats were humanely euthanized with an overdose of sodium pentobarbital. Rats were randomized to SCI and control groups. It was not possible to blind the surgeon performing the craniectomy and neurophysiological data collection to group assignment, but these data were collected using an automated system. The data analyst performing the post-hoc spike discrimination procedures for each animal's neurophysiological data was blind to group assignment.

**Statistics**. Statistics were analyzed using Graphpad Prism 6. For spine density measurements, we used paired two-tailed $t$-tests. Sample sizes were determined by power analysis, setting a desired Cohen's d statistical power to 0.8 and using a 5% error rate (type I). For the paired spine density analyses (Figs. 1d, 6b, c, and 8b, i), we calculated a minimum sample size of 10 primary dendritic processes per condition using means and s.d.'s from Fig. 1d. For FM unloading analyses (Figs. 3c, i, 6e, g, Supplementary Fig. 3b–f), we used a minimum sample size of 50 FM puncta per condition based on previously published data[24] and using means and s.d.'s from Fig. 3c. For unresponsive/responsive FM puncta analyses (Figs. 3d, e, 6d, f, and 8c, d), we calculated a minimum sample size of 6 frames per condition using means and s.d.'s of unresponsive puncta in uninjured control vs. 48 h post-axotomy (Fig. 3e). For vGAT/vGLUT1 puncta per area analyses (Figs. 5c and 8e, f, j), we calculated a minimum sample size of 8 neuron fields per condition using the means and s.d.'s of vGAT puncta per area in control and axotomized samples

(Fig. 5c). For the fraction of GAD67 or vGLUT1 puncta to FM puncta analyses (Fig. 5a, b), we calculated a minimum sample size of 18 frames per condition using the means and s.d.'s found in Fig. 5b. For DCC immunofluorescence per spine analyses (Fig. 8h), we calculated a minimum of 75 samples per condition using the means and s.d.'s of controls and axotomized samples. We estimated requisite samples size for recordings of spontaneous postsynaptic currents by assuming a doubling or halving of mEPSC or mIPSC parameters following axotomy in correlation with the magnitude of changes observed in our analysis of spine numbers, presynaptic release data, and based on similar effect sizes being reported following synaptic activity blockade[64, 65]. s.d. values for the power analysis were estimated based on previously published values from dissociated hippocampal neuron electrophysiological recordings[64, 65]. This analysis suggested we required 13 samples per group for mEPSCs and 9 samples per group for mIPSCs. All recording, (Figs. 3g, h and 5d–f) meet or exceed the sample size required by our power analysis. For in vivo experiments, hypotheses regarding spike firing rates were tested independently in each cortical layer (Va, Vb, and VI) using a two-tailed $t$-test, ($\alpha = 0.05$). Samples were not excluded from our data sets and sample size was determined based on our experience and previous publications[24]. As confirmation of sufficient sample sizes, we used the means and s.d.'s for layer Vb control and SCI conditions and calculated a minimum of 112 samples for control and 282 samples for SCI conditions. For all experiments, sample sizes were equal or greater than the calculated minimum sample sizes.

**Data availability**. The microarray data set generated during the current study is available in the Gene Expression Omnibus (GEO) repository, www.ncbi.nlm.nih.gov/geo/. The GEO accession number is GSE89407. All relevant data will be available from authors upon request. The spinal cord injury microarray data set analyzed in this study was generated and published by a research group independent from the authors; the raw data is available at EMBL-EBI Array Express, www.ebi.ac.uk/arrayexpress/, accession number E-MTAB-794[37].

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

## Acknowledgements

We thank Stephanie Gupton for netrin-1, Kelly Carstens for preliminary gene expression work, Cassie Meeker for technical support, and Fabio Urbina for assistance in statistical analysis. We thank Richard Segal (MUSC), Julius Dewald (RIC), and Taylor lab members for their advice and discussions. A.M.T. acknowledges support from the American Heart Association (17GRNT33700108), Eunice Kennedy Shriver NICHD (K12 HD073945), NIMH (R42 MH097377), and an Alfred P. Sloan Research Fellowship. Imaging was partially performed at the Neuroscience Center Microscopy Core Facility, supported, in part, by funding from the NIH-NINDS Neuroscience Center Support Grant P30 NS045892 and the NIH-NICHD Intellectual and Developmental Disabilities Research Center Support Grant U54 D079124. R.S.L was supported by NRSA predoctoral fellowship F31 MH091817 and the UNC Department of Cell Biology and Physiology's Dr Susan Fellner fellowship. R.L.B. was supported in part by NIGMS grant 5T32 GM007092. B.D.P was supported by NINDS grant R01 NS085093. R.J.N. was supported by NINDS grant R01 NS30853 and the Ronald D. Deffenbaugh Family Foundation.

## Author contributions

T.N. designed and performed experiments and wrote the manuscript. R.S.L. designed and performed electrophysiology experiments and wrote portions of the manuscript. R.L.B. designed and performed experiments. S.B.F. performed experiments. B.D.P. designed

experiments. R.J.N. designed and performed experiments. A.M.T. designed experiments, analyzed data, and wrote the manuscript.

## Additional information

**Competing interests:** A.M.T. is an inventor of the microfluidic chambers (US 7419822 B2) and has financial interest in Xona Microfluidics, L.L.C. The remaining authors declare no competing financial interests.

