## [Peer Review File · Nature Communications]

Reviewers' comments:

Reviewer #1 (Remarks to the Author):

This study aimed to evaluate synaptic responses to axotomy in a microfluidic chamber where the somatodendritic and the axonal compartment of pyramidal neurons are physically separated. The main findings are that in the days following the axonal injury, dendritic spines are lost followed by increased synaptic vesicle release onto the injured neurons. This was due to a selective removal of silent inhibitory spine synapses. The authors went on to show that the decrease in silent puncta after axotomy was dependent on gene transcription and independent of activity and that netrin 1 can rescue inhibitory puncta loss. The authors claim that this is a new in vitro model to study neuron intrinsic synaptic properties following axonal injury relevant to stroke and traumatic brain injury.

The study of intrinsic responses to injury is timely and important and the authors have made a nice addition to the field by showing the novel role of netrin-1 in selective presynaptic inhibitory puncta loss following axotomy. However, the in vivo relevance of these results should be addressed.

While the microfluidic chamber is a great system to analyse axon-specific RNA and protein repertoire changes after axotomy and for drug discovery, this is not what the authors have done here. In fact, it is not clear to this reviewer what is novel about the in vitro microfluidic system used here compared to the 2005 Nature methods paper from the same author (AMT), where axon injury was shown as one of the applications of the microfluidic chamber.

There are other issues which should be addressed before publication:

- The first is why the authors have used hippocampal neuron cultures to model cortical injury, as opposed to cortical cultures, which are also widely used.
- The authors show that silent puncta loss after axotomy depends on gene transcription, can the authors show that also spine loss depends on transcription and that netrin1 applications rescue the dendritic spine loss?
- A key experiment is missing, i.e. to downregulate netrin-1/DCC to see if this is sufficient to cause the structural changes in dendritic spines and presynaptic inhibitory terminals.
- How does the firing rate of injured neurons change at 24hr, 48hr?
- Line 124-125 'These results were similar to those obtained by examining the entire field of FM puncta rather than selecting only puncta that colocalize with eGFP expressing neurons', this is unexpected considering the results in fig 2h and 2i.
- How does the specific reduction of silent puncta take place mechanistically?
- Line 176: 'thus we next asked whether the eliminated puncta were primarily GABAergic or

glutamatergic', how did the authors selectively study the eliminated silent puncta?

- Line 234: The focus on Netrin-1 as opposed to the other 5 transcripts which were also significantly changed after axotomy needs to be justified.

- Line 236: 'axotomized and uninjured neurons harvested from microfluidic chambers', is it possible to do this analysis on injured vs uninjured neurons, rather than different chambers?

- The timing of the netrin-1 application needs justification, why 40hr post-axotomy?

- Previous work on the role of Nitric Oxide in synaptic remodelling (Sunico et al J. Neurosci. 2005) and in vivo single neuron axotomy (Canty et al J. Neurosci. 2013) should also be discussed.

- Fig. 2g-i: the mEPSC frequencies after axotomy are not different between 2g (injured+uninjured) and 2h (injured only), how can the authors explain this finding?

Minor points:

lines 167-170: This sentence is confusing and needs rephrasing

line 637: why was a one-tailed test used?

lines 118-120: enhanced synaptic vesicle release rate persisted until 4 days, why are the data not shown?

line 140: 'delayed enhancement in the fraction of responsive presynaptic terminals', this sentence is at odds with the finding that there is no effect on responsive terminals but a decrease in the number of silent FM puncta.

Figure 2g-I: to improve clarity it would be good to add an x-axis title to indicate the time point

lines 178-182: these data are not shown in a figure

line 78: add 'expressing fluorescent proteins' after G-deleted rabies virus

line 99: 1e not 2e

line 110: why eGFP and not mCherry as in the previous experiments?

line 165: 2i not 2j

line 197: glutamatergic not glutamergic

line 206: 'changes in the proportion of responsive', incomplete sentence.

Line 278: influence not influences

Fig.1d: not clear that there is 25% loss in thin spines after one day, as there are hardly any thin spines in these images. Also in the graph 'before' is misleading for the control experiments.

Fig.1e: Spine numbers need to be expressed per unit length (i.e. density) or area to understand the magnitude of the changes

Fig.6b: add lower magnification to evaluate the staining distribution better

Reviewer #2 (Remarks to the Author):

In this MS the Authors study the effects of axotomy on neuronal excitability in the 48 hours that follow trauma. This is an in vitro study that exploits a smart microfluidic chamber that allows to separate the somatic and axonal compartments for physiological and biochemical studies. Their main findings are that the number of spines is decreased (as already shown in a few in vivo studies), and that excitability is altered by means of a mix of pre and post synaptic mechanisms. Finally, they identify netrin1 as a factor regulating these processes. Most of these findings are novel and quite interesting.

The study addresses a problem of great importance by means of an impressive mix of different and complementary techniques. The data are of very good quality and I don't have any major criticism. However, I not always agree with the data interpretation and I have several questions that I would like to see answered.

1) Unless I have not properly understood the experimental set-up, I have a basic concern about the experimental model. Here, axotomized neurons are different from control neurons, that grow their axons within the somato-dendritic compartment of the chamber, since they do not form any synaptic connection. Given the role of retrograde synaptic signaling on neuron physiology, biochemistry and control of gene expression, I wonder whether the responses to axotomy would be different if the neurons elongating in the micro-channels would find synaptic targets in the axon terminal compartments. Would it be possible to seed neurons also in the axonal compartment in order to provide proper synaptic targets to the soon-to-be-cut axons?

2) The spine loss experiment should be interpreted more carefully. The early event is a decrement in thin spines. Early in development, filopodia are the precursors of properly connected spines and typically they have an extremely fast turnover, appearing and disappearing in matter of tenth of minutes. A snapshot will only provide the steady state balance between these processes. A possible interpretation is that the axotomized neurons actively reduces the production of novel filopodia without affecting their removal. Personally, I would love to see a time lapse experiment in which motility and turnover of filopodia is assessed 24 hours after axotomy. The reduction in large mushroom spines at later time point might be due, at least in part, to a hindered stabilization of filopodia into mushroom spines. Indeed, the spine loss after 24 hrs indicated in fig 1C is not that different from the new spines that would have appeared due to normal synaptogenesis (about twice as much as what is observed in the left panel of fig 1C. If this is the case, what has been observed is NOT accounted for by spine loss but it is due to the failure of forming new spines. Actually this point is very interesting and it is worth been properly assessed also because it might suggest that he mechanisms at play in young neurons (this culture system) might be different from the mechanisms that would be at work in the mature brain.

3) Page 7: "Our results suggest that distal axotomy triggers a retrograde cascade leading to enhanced presynaptic excitability". I agree only partially with this statements. Data can be interpreted slightly differently: if the spine loss would occur mainly to silent synapses, we would still see a net increase of excitability through a post synaptic mechanisms of selective spine elimination. Indeed, The increase in size of the mEPSC showed in Fig 2h can be due either to the selective elimination of silent or quasi-silent spines, or to the selective

enrichment of AMPAR of the remaining spines. To summarize, figure 2 indicates that both pre and post-synaptic mechanisms are into play.

4) It would be very interesting to perform the immunohistochemistry for PSD95 to see on an individual spine basis whether the excitability increase would come with an enhancement of the post-synaptic density. On a similar note, it would be interesting to see the statistics of the spine head volume before and after axotomy. This can be estimated by the integral of the spine head fluorescence normalized for the nearby dendrite fluorescence. Depending on the characteristics of the imaging data acquired for the study, all the necessary data might already be there and would "only" require proper analysis.

5) The data about GABAergic synapses is quite interesting and the immune data are pretty convincing. Here I have a very basic issue that should somehow be mentioned. At the time point of the study (13 DIV) the intracellular Cl concentration should still be immature in a fraction of neurons, therefore, GABA currents might be depolarizing. As I mentioned before (point 2), what might be happening here might not be completely representative of what would happen in the mature brain. This point would be put to rest with the analysis of mIPSC (similarly to what done for mEPSC) by means of perforated patch (so that not to alter the Cl content of the cell). These experiments would also allow to estimate the Cl reversal, which could be quite interesting...

6) One missing aspects of this study is the lack of a mechanistic interpretation between the axotomy and the observed changes. Axotomy is known to activate several signaling mechanisms (well described by Rishal and Fainzilber, 2014) occurring at different time points after the lesion. The earliest signaling mechanism, compatible with the timing of the transcription-block experiments, include a retrograde propagating calcium wave which depends on both the activation of NaV channels and on the reversed operation of the NaCa exchange. Surprisingly, the TTX experiments seem to rule out the involvement of retrograde firing on signaling from the lesion site to the cell bodies. However, depending on the details of the axotomy process, the exchanger could be inverted if enough Na would leak in at the lesion site. This could still trigger the formation of a Ca wave that would back-propagate along the axon. It would be interesting to observe whether the blockage of the NaCa exchange at the lesion time (with Bepredil, for example) would have an effect on the response to axotomy.

Minor points:

1) The axotomy process is of obvious crucial importance in this study. Therefore some details should be provided in the Methods even if the technique has been described in details elsewhere.

2) Figure 1d. The quality of the spine images is poor. It might well be a problem of the PDF I have at hand. It might be useful to process the image with a non-linear stretching in order to show properly the fainter details (like thin spines) without oversaturating the larger spine heads. The use of a non linear stretching should be disclosed in the figure legend.

3) Figure 2g. Panels h and I are far more informative and so panel g feels a bit pleonastic.

4) There are a few repetitions in the Method section that requires some careful reading and editing. In general, I found that the Methods are really a bit too succinct.

Reviewer #3 (Remarks to the Author):

In this paper, Nagendran and colleagues examined the impact of axotomy on the innervation of damaged neurons using an in vitro model system based on the use of microfluid chambers. This system allows the compartmentalization of the end of axons (a compartment where they can be axotomised), the axon shaft and the somatodendritic compartment where changes in synapses onto the injured and uninjured neurons could be monitored. Specific dyes or viral infection on the axonal compartment allows the specific labelling and therefore identification of injured neurons (cell bodies and dendrites) on the somatodendritic compartment. This is a nice system of identifying retrograde and trans-synaptic changes.

Like previously reported, the authors showed that axotomy leads to the loss of dendritic spines on injured neurons. This is also accompanied by the loss of GABA inhibition as determined by the decrease in the number of vGAT puncta in association with remaining spines. Indeed, the loss of GABA inhibition has been proposed to lead to increased excitability. The authors examined here the mechanisms that contribute to this loss of inhibition by showing that Netrin is involved.

To evaluate presynaptic changes FM dyes were used. The authors found an increase in the unloading rate of FM suggesting increased release onto the dendrites of injured cells. The authors also report that patch clamp recordings of injured neurons reveal an increase in the frequency and amplitude of mEPSCs. The increased frequency is consistent with their finding with FM dye. The authors proposed that axotomy leads to the loss of vGat puncta onto a subset of dendritic spines resulting in increased excitability. To explain the increased FM labelling, the authors proposed that a trans-synaptic signal (Netrin) is involved. However, some of the results do not seem to come together.

The authors made a strong case that trans-synaptic signalling from the injured neuron results in a decreased inhibition by GABA, due to the loss of GABA inputs, results in increased excitability. However, recordings of mIPSCs are missing. Importantly, many of quantifications for FM puncta or innervation (vGlut and vGat puncta) were normalized so it is difficult to assess the effect.

The authors also showed that TTX, which blocks action potentials, does not affect the effects induced by axotomy. TTX should affect both excitatory and inhibitory synapses. One would presume that TTX would affect inhibitory inputs and therefore block some of the effect of axotomy. The results are not explained.

Finally, the microarray data is poor. The authors showed two figures with just a number of dots representing the different genes regulated following axotomy but they only labelled

one (netrin). They only mentioned that Netrin is affected after axotomy as data not shown. Importantly, the table containing the list of genes identified in their screen is missing! The table only has the title.

In conclusion, the authors present a number of very interesting observations including the finding that a retrograde signal from injured axons influences specific inputs. However, there are concerns about some of the data and conclusions. Based on these concerns, this paper is not appropriate for Nature Communication in its current form.

Specific comments:

- 1) In Figure 1, the authors showed that processes grow from the somato-dendritic compartment into the grooves. However, the authors did not demonstrate that these processes are axons. Proper staining for axonal and dendritic markers should be used to demonstrate the specificity of the compartmentalization.
- 2) Figure 1d: The data on spine density should not be normalized to 1 but presented as spines per unit length of dendrite.
- 3) Figure 1d: the data on synapse density after 24 hour axotomy does not coincide with the data presented in Figure suppl 2. Here there is a 25% decrease, in supplementary figure 2 there is a 70% decrease. The authors need to explain this discrepancy.
- 4) Figure 2 and page 7, first paragraph. The authors indicated that after 48 hour axotomy there is an increase in the number of responsive FM puncta although the total number does not change. The conclusion is that there is shift from silent to responsive FM puncta. This finding poses a question. If the number of spines decreases but the total number of FM puncta onto the injured cell does not, then are some of these puncta orphan sites? If so, how do they contribute to excitability?
- 5) In Figure 2, it is not clear whether the FM puncta that change are those in apposition to the injured cell. This is crucial to determine how specific the labeling using the FM dye is.
- 6) Figure 3 shows normalized data on the fraction of vglut/spines. This data, like in other figures, should be shown as number of vglut puncta along a given length of dendrite.
- 7) Figure 4. The authors used TTX to examine its impact on the response after axotomy and reported that TTX does not affect the axotomy-mediated response on FM. This is puzzling because TTX should affect both inhibitory and excitatory synapses. So inhibitory terminals should be silenced. A possible conclusion is that the loss of inhibitory inputs is not relevant to increase presynaptic response. The authors should reconcile these apparently conflicting results.
- 8) Figure 5, it is surprising that the authors did not reveal the identity of the genes in this graph other than netrin.
- 9) Table Supplement 2, has a title that says "List of transcripts that are significantly changed 24 h after injury". However, the table is empty. Any explanation?
- 10) Figure 6, the number of DCC puncta in association with the injured neuron and also on non-injured neuron should be presented.
- 11) Figure 6g seems to show that Netrin rescues the decrease in GAD67 puncta after axotomy but there is a clear tendency to decrease. The analysis was done on 7 chambers. Given that no details are given about how many experiments these 7 chambers come from,

the authors should provide clear evidence that this effect or lack of effect is real. For example, they should also analyse the number of vGAT puncta.

Reviewers' comments (bold, italicized):

Reviewer #1 (Remarks to the Author):

This study aimed to evaluate synaptic responses to axotomy in a microfluidic chamber where the somatodendritic and the axonal compartment of pyramidal neurons are physically separated. The main findings are that in the days following the axonal injury, dendritic spines are lost followed by increased synaptic vesicle release onto the injured neurons. This was due to a selective removal of silent inhibitory spine synapses. The authors went on to show that the decrease in silent puncta after axotomy was dependent on gene transcription and independent of activity and that netrin 1 can rescue inhibitory puncta loss. The authors claim that this is a new in vitro model to study neuron intrinsic synaptic properties following axonal injury relevant to stroke and traumatic brain injury.

1. The study of intrinsic responses to injury is timely and important and the authors have made a nice addition to the field by showing the novel role of netrin-1 in selective presynaptic inhibitory puncta loss following axotomy. However, the in vivo relevance of these results should be addressed.

We thank the reviewer for raising this important point. We now include new evidence that further establishes the importance of netrin-1 in an *in vivo* nerve injury model. Previous studies have examined cortical gene expression from retrograde labeled layer V rat cortex either given sham injury or spinal cord hemisection at thoracic level 8 at both 1 and 7 days following injury (Jaerve et al., Plos One, 2012; PMID: 23236355). We analyzed the raw data from this paper, downloaded from EMBL-EBI Array Express, and our new analysis shows that spinal cord injury reduces netrin-1 expression in cortical layer V by 7 days ($p < 0.05$, 2-way ANOVA, Sidak's multiple comparisons test), consistent with our model. These data provide direct, *in vivo* evidence that netrin-1 levels change with injury, and offer an independent validation of the link between netrin-1 and nerve injury. In combination with the extensive new data prepared for our re-submission, we are confident that our new analysis highlights the *in vivo* relevance of altered netrin signaling in nerve injury (now presented in Figure 7). In addition, netrin family proteins have been shown to be downregulated following brain injury in adult rats and have been shown to improve recovery outcomes; our manuscript now cites these important publications that were regrettably absent from our initial submission (Manitt et al. 2006; Ahn et al. 2007; Lu et al. 2011; Sun et al. 2011; Han et al. 2016). These points are now discussed in Results, section entitled, 'Netrin-1 mRNA down-regulated post-axotomy' on page 15.

We now present additional *in vivo* data in collaboration with Prof. Randy Nudo's lab, which further supports our findings and the relevance of our data. In our microfluidics-based model we found that directly injured pyramidal neurons show enhanced excitability following axotomy compared with neighboring uninjured neurons. We wanted to determine whether directly injured long projection corticospinal neurons within cortical layer V subjected to distal nerve injury would similarly show a preferential enhancement in excitability. Using a thoracic spinal cord contusion model in rats, we measured spiking frequency in hindlimb motor cortex layer Vb, containing the highest density of de-efferented corticospinal neurons, and compared these frequency values with adjacent cortical layers Va and VI. Spiking frequency was measured between 4 and 18 weeks post-injury. Importantly, we found that the spiking rate of neurons within layer Vb of hindlimb motor cortex was significantly elevated compared

with neighboring layers Va and VI. These important new data are added as new **Figure 4** which is discussed in Results, section entitled 'Spinal cord injury (SCI) induces persistent and enhanced firing rates in layer Vb', page 10.

2. While the microfluidic chamber is a great system to analyse axon-specific RNA and protein repertoire changes after axotomy and for drug discovery, this is not what the authors have done here. In fact, it is not clear to this reviewer what is novel about the in vitro microfluidic system used here compared to the 2005 Nature methods paper from the same author (AMT), where axon injury was shown as one of the applications of the microfluidic chamber.

The reviewer is correct, the chambers used in this manuscript are the same as previously described in the Nature Methods paper. We have now removed the word "novel" from the abstract, introduction, and discussion in reference to the chambers. While the chambers are the same as previously described, we believe the experimental *use* of these chambers to examine retrograde plasticity following axotomy is novel.

There are other issues which should be addressed before publication:

3. The first is why the authors have used hippocampal neuron cultures to model cortical injury, as opposed to cortical cultures, which are also widely used.

We used hippocampal cultures because pyramidal neurons account for roughly 85-90% of the cells making up the hippocampus, while the cortex is much more heterogeneous in cell type (Banker and Goslin 1998). Embryonic hippocampal cultures also have consistent numbers of GABAergic interneurons (~10% of the neuron population) as found *in vivo*. These characteristics make hippocampal cultures more consistent batch-to-batch than cortical cultures. Further, pyramidal cell morphology of hippocampal cultures is comparable to cell morphology found *in vivo* in the maturing brain (Banker and Cowan 1977). Text was added to clarify these points in Results, section entitled, 'In vitro model to study distal axon injury to pyramidal neurons' page 4.

In addition, to test whether cortical neurons show similar synaptic changes after axotomy, we cultured cortical neurons within microfluidic chambers and performed FM unloading experiments as described in **Figure 3**. As background, we found a robust difference in FM unloading at presynaptic terminals in response to field stimulation between axotomized and uninjured control chambers with hippocampal neurons. We wanted to confirm a similar difference in the rate of unloading would occur in cortical cultures. **Figure supplement 4** summarizes the results of this experiment, showing that the cortical neurons respond similarly to hippocampal neurons following injury. Again, we used hippocampal neurons because they provide a more consistent neuron population batch-to-batch and are enriched for pyramidal neurons. We updated the text to describe this important control experiment [Results, section entitled 'A persistent enhancement in synaptic vesicle release rate follows distal axon injury', 2nd paragraph, page 7].

4. The authors show that silent puncta loss after axotomy depends on gene transcription, can the authors show that also spine loss depends on transcription and that netrin-1 applications rescue the dendritic spine loss?

We thank the reviewer for suggesting these experiments which we have now performed and believe have greatly strengthened our findings. First, we used the reversible transcription inhibitor, DRB, applied selectively to the somatic compartment 1h during axotomy and quantified spine density in axotomized and control uninjured chambers. As expected, we found that spine density measured 24 h after DRB treatment was not statistically different than control uninjured neurons receiving DRB. These results show that when transcription is inhibited at the time of injury, the reduction in spine density caused by axotomy is prevented 24 h post-axotomy. These results are now included in new **Figure 6c** and discussed in the Results, section entitled 'Local activity and differential gene expression regulate axotomy induced synaptic changes', 2nd paragraph, pages 13-14.

We also quantified spine density before and 2 days after axotomy in the presence of netrin-1 applied for 8 h at 40 hours following axotomy. As we hypothesized, we found that netrin-1 normalized the spine density at 48 h post-axotomy to levels at or greater than before axotomy. These new data and results are presented in **Figure 8a,b** [Results, section entitled, 'Exogenous netrin-1 normalizes injury-induced changes in spine density and presynaptic release', pages 15-16].

5. A key experiment is missing, i.e. to downregulate netrin-1/DCC to see if this is sufficient to cause the structural changes in dendritic spines and presynaptic inhibitory terminals.

We thank the reviewer for suggesting this experiment and have now added this important new data to our revised manuscript. We used a function blocking antibody for the netrin-1 receptor, DCC, to downregulate netrin-1 signaling within the somatodendritic compartment of uninjured neurons and found that spine density was significantly reduced 24 h following initiation of treatment compared with IgG control treatment (**Figure 8i**). [Results, section entitled, 'Exogenous netrin-1 normalizes injury-induced changes in spine density and presynaptic release', 3rd paragraph, pages 16-17].

To determine whether blocking netrin-1/DCC signaling would be sufficient to cause loss of presynaptic inhibitory terminals, we quantified the number of inhibitory (vGAT) presynaptic terminals following 24 h of treatment with the DCC function blocking antibody compared to control IgG. Supporting the role of netrin-1/DCC in regulating inhibitory terminal loss, we found that vGAT positive terminals colocalizing with retrograde labeled pyramidal neurons were significantly reduced in the presence of DCC antibody (**Figure 8j**). Somewhat surprisingly, we found that there was no reduction in the number of vGLUT1 positive terminals colocalizing with labeled neurons as other groups found when blocking DCC (Goldman et al. 2013). There are a couple potential explanations for our data. First, we used a shorter treatment time with the DCC antibody than previously published reports. Also, our analysis examined fields containing mostly somata and proximal dendrites where a higher fraction of inhibitory terminals form synapses. [Results, section entitled, 'Exogenous netrin-1 normalizes injury-induced changes in spine density and presynaptic release', 3rd paragraph, pages 16-17; Discussion, 6th paragraph, page 20].

6. How does the firing rate of injured neurons change at 24hr, 48hr?

Our new *in vivo* results show that firing rate is significantly increased in de-efferented layer Vb motor cortical neurons between 4-18 weeks post injury (new **Figure 4**). Due to the

addition of this new *in vivo* data, we did not pursue experiments examining the firing rate post-axotomy in cultured neurons.

7. Line 124-125 'These results were similar to those obtained by examining the entire field of FM puncta rather than selecting only puncta that colocalize with eGFP expressing neurons', this is unexpected considering the results in fig 2h and 2i.

We recorded fields of view nearest the barrier region where a large fraction of axotomized neurons are located. Thus, a significant proportion of labeled presynaptic terminals formed onto these axotomized neurons. Comparing the FM unloading curves from entire fields versus FM puncta that colocalize with eGFP labeled neurons, the entire field data shows less pronounced, but still significant, differences in time constant values. We have also found that regions far away from axotomized neurons in the somatic compartment show unloading comparable to uninjured controls, further supporting that the directly injured neurons mediate trans-synaptic changes in release at the time-points examined; these new data are presented in **Figure 3i** and discussed further in comments to *reviewer #3, comment 8* and in Results, section entitled 'Enhanced glutamate release occurs at synapses onto injured neurons, 2nd paragraph, page 9.

8. How does the specific reduction of silent puncta take place mechanistically?

First, we believe our use of the word "silent" in the original submission may have been misleading because silent synapses have been defined as synapses without postsynaptic AMPA receptors. Since we examined the responsiveness of FM dye-stained synaptic vesicles to unloading to an electrical stimulus, we now refer to the FM puncta that do not unload dye in response to electrical stimulation as "unresponsive" rather than silent.

Because we used a strong depolarization agent (KCl) we found that the vast majority of presynaptic terminals (both excitatory and inhibitory) were labeled (**Figure supplement 3**); thus the reduction in the number of FM puncta post-axotomy that we observed suggests overall fewer presynaptic terminals onto axotomized neurons 48h post-axotomy which fits with the observed reduction in spine density (now **Figure 3e**). The increase in the fraction of responsive puncta suggest the remaining presynaptic terminals were more excitable. Our mEPSC and new mIPSC data support our FM data, as we found increases in the frequency of spontaneous release from both excitatory and inhibitory terminals (now **Figure 3** and new **Figure 5**, respectively). The enhanced release rate at preserved terminals 48 h post-axotomy occurred following postsynaptic changes, suggesting synaptic homeostasis involving retrograde communication. The Results and Discussion have been revised to clarify these points.

9. Line 176: 'thus we next asked whether the eliminated puncta were primarily GABAergic or glutamatergic', how did the authors selectively study the eliminated silent puncta?

We have revised our wording and we no longer refer to unresponsive FM puncta as silent. We rephrased the wording in this section to read, "Loss of inhibition following distal injury contributes to enhance excitability *in vivo*, thus we wanted to test whether axotomy in our culture system results in a similar loss of inhibitory terminals." Results, section entitled, 'Axotomy selectively eliminates GABAergic terminals onto spines of injured neurons', page 11.

10. Line 234: The focus on Netrin-1 as opposed to the other 5 transcripts which were also significantly changed after axotomy needs to be justified.

We thank the reviewer for bringing up this point. Netrin-1 signaling has been shown in numerous studies to be downregulated following injury and recent studies have shown that netrin-1 improves recovery following injury; these citations are given in more detail in *reviewer #1, comment #1*. Further, our new analysis of *in vivo* gene expression data shows that netrin-1 is significantly downregulated in cortex of young adult rats by 7 days following spinal cord hemi-transection (new **Figure 7**).

11. Line 236: 'axotomized and uninjured neurons harvested from microfluidic chambers', is it possible to do this analysis on injured vs uninjured neurons, rather than different chambers?

The reviewer brings up a great point. While this experiment is beyond the scope of this paper, it is something we would like to do in the future using laser capture microdissection to collect directly injured neurons and adjacent uninjured neurons. We would expect to see higher fold-changes in gene expression between directly injured and uninjured neurons than in our current microarray data which used material harvested from all cells within the somatodendritic compartment.

12. The timing of the netrin-1 application needs justification, why 40hr post-axotomy?

We applied netrin-1 1 ½ days after axotomy when spine changes were observed and applied it for 8 h because we wanted to observe stable synaptic changes that may not occur with shorter treatment times. Netrin-1 has been applied for 1-2 days at a similar concentration without deleterious effects (Menon et al. 2015). We added this explanation to the text [Methods, section entitled 'Drug treatments', page 28-29].

13. Previous work on the role of Nitric Oxide in synaptic remodelling (Sunico et al J. Neurosci. 2005) and in vivo single neuron axotomy (Canty et al J. Neurosci. 2013) should also be discussed.

The Sunico et al. paper is now been discussed [Discussion, 5th paragraph] and Canty et al paper is cited [Discussion, 1st paragraph].

14. Fig. 2g-i: the mEPSC frequencies after axotomy are not different between 2g (injured+uninjured) and 2h (injured only), how can the authors explain this finding?

The discrepancies in these previous graphs were due to sample size differences. We performed more recordings and replaced these data to clarify our results. These new data are shown in **Figure 3g,h**. In new Figure 3g we analyzed mEPSC frequency and amplitude for directly axotomized neurons compared to uninjured neurons in control chambers. In new Figure 3h we examined axotomized or "cut" neurons and compared these recordings to neighboring "uncut" neurons within the same axotomized chambers. We found that the "cut" neurons show a significant increase in mEPSC frequency whereas the neighboring "uncut" neurons do not. These data are now described in Results, section entitled, 'Enhanced glutamate release occurs at synapses onto injured neurons, pages 8-9.

Minor points:

lines 167-170: This sentence is confusing and needs rephrasing

We rephrased this sentence to “Further, directly injured neurons trans-synaptically influenced presynaptic glutamate release without affecting nearby synapses at uninjured neurons.” [Results, section entitled, ‘Enhanced glutamate release occurs at synapses onto injured neurons, 2nd paragraph, page 9.]

line 637: why was a one-tailed test used?

Because we tested whether means were different (either higher or lower), a two-tailed test was, in fact, more appropriate. In the revised manuscript, we have used two-tailed tests throughout, along with more data provided by the additional experiments. Our new data is shown in **Figure 3g**.

lines 118-120: enhanced synaptic vesicle release rate persisted until 4 days, why are the data not shown?

We have now included this data in **Figure supplement 4**. Four (4) days after axotomy we found that within the entire field close to the barrier region, where a large percentage of neurons are axotomized, there was a significantly increased presynaptic release rate, though a more moderate difference than at 48 h post-axotomy [Results, section entitled ‘A persistent enhancement in synaptic vesicle release rate follows distal axon injury’, 2nd paragraph, pages 7-8].

line 140: 'delayed enhancement in the fraction of responsive presynaptic terminals', this sentence is at odds with the finding that there is no effect on responsive terminals but a decrease in the number of silent FM puncta.

We have revised our text to be clearer. The fraction of responsive terminals increased 48 h post-axotomy, though the density of spines and FM terminals decreased following axotomy. We have clarified these points in the text and no longer discuss the absolute levels of responsive terminals which, we agree, is a confusing point. [Results, section entitled ‘A persistent enhancement in synaptic vesicle release rate follows distal axon injury’, 3rd paragraph, page 8].

Figure 2g-l: to improve clarity it would be good to add an x-axis title to indicate the time point

An x-axis title has been added to read “48 h post-axotomy”. These panels are now **Figure 3g,h**.

lines 178-182: these data are not shown in a figure

We now show these data quantifying the fraction of vGLUT1 or GAD67-positive FM puncta at 48 h post-axotomy in **Figure 5a,b**.

line 78: add 'expressing fluorescent proteins' after G-deleted rabies virus

This has been added. [Results section, 1st paragraph, page 5]

line 99: 1e not 2e

We separated this data from Figure 1 and it is now shown in Figure 2.

line 110: why eGFP and not mCherry as in the previous experiments?

eGFP was used because the fluorescence spectra of the FM dyes we used overlaps with mCherry expression.

line 165: 2i not 2j

These data are now shown in Figure 3.

line 197: glutamatergic not glutamergic

This has been corrected.

line 206: 'changes in the proportion of responsive', incomplete sentence.

This has been corrected.

Line 278: influence not influences

This has been corrected.

Fig.1d: not clear that there is 25% loss in thin spines after one day, as there are hardly any thin spines in these images. Also in the graph 'before' is misleading for the control experiments.

We have replaced the representative images with ones that we feel are more representative of our data (new **Figure 2b**). We also relabeled the uninjured control graph in new **Figure 1c** with "0 h / Before" instead of "Before".

Fig.1e: Spine numbers need to be expressed per unit length (i.e. density) or area to understand the magnitude of the changes

We have replaced the graphs so that spine density is quantified (**Figures 1, 2, 6, 8**).

Fig.6b: add lower magnification to evaluate the staining distribution better

We replaced these images and they are now shown in **Figure 8g**.

Reviewer #2 (Remarks to the Author):

In this MS the Authors study the effects of axotomy on neuronal excitability in the 48 hours that follow trauma. This is an in vitro study that exploits a smart microfluidic chamber that allows to separate the somatic and axonal compartments for physiological and biochemical studies. Their main findings are that the number of spines is decreased (as already shown in a few in vivo studies), and that excitability is altered by means of a mix of pre and post synaptic mechanisms. Finally, they identify netrin1 as a factor regulating these processes. Most of these findings are novel and quite interesting.

The study addresses a problem of great importance by means of an impressive mix of different and complementary techniques. The data are of very good quality and I don't have any major criticism. However, I not always agree with the data interpretation and I have several questions that I would like to see answered.

1) Unless I have not properly understood the experimental set-up, I have a basic concern about the experimental model. Here, axotomized neurons are different from control

neurons, that grow their axons within the somato-dendritic compartment of the chamber, since they do not form any synaptic connection. Given the role of retrograde synaptic signaling on neuron physiology, biochemistry and control of gene expression, I wonder whether the responses to axotomy would be different if the neurons elongating in the micro-channels would find synaptic targets in the axon terminal compartments. Would it be possible to seed neurons also in the axonal compartment in order to provide proper synaptic targets to the soon-to-be-cut axons?

The reviewer raises an excellent point. To test whether axotomized neurons that have functional synaptic connections show similar retrograde synaptic changes as with untargeted axons, we performed additional FM experiments with target neurons added to the axonal compartment. By adding this target population, axons were able to form connections as we have shown and characterized in previous work (Taylor et al. 2010). We then axotomized cultures by aspirating out axons together with target neurons and then compared FM unloading curves 48 h post-axotomy. We found equivalent differences in the rate of unloading between axotomized cultures with and without synaptic targets. We added text to describe this control experiment and these new data are shown in **Supplemental Figure 4** [Results, section entitled 'A persistent enhancement in synaptic vesicle release rate follows distal axon injury', 2nd paragraph, page 7-8].

2) The spine loss experiment should be interpreted more carefully. The early event is a decrement in thin spines. Early in development, filopodia are the precursors of properly connected spines and typically they have an extremely fast turnover, appearing and disappearing in matter of tenth of minutes. A snapshot will only provide the steady state balance between these processes. A possible interpretation is that the axotomized neurons actively reduces the production of novel filopodia without affecting their removal. Personally, I would love to see a time lapse experiment in which motility and turnover of filopodia is assessed 24 hours after axotomy. The reduction in large mushroom spines at later time point might be due, at least in part, to a hindered stabilization of filopodia into mushroom spines. Indeed, the spine loss after 24 hrs indicated in fig 1C is not that different from the new spines that would have appeared due to normal synaptogenesis (about twice as much as what is observed in the left panel of fig 1C. If this is the case, what has been observed is NOT accounted for by spine loss but it is due to the failure of forming new spines. Actually this point is very interesting and it is worth been properly assessed also because it might suggest that he mechanisms at play in young neurons (this culture system) might be different from the mechanisms that would be at work in the mature brain.

The reviewer raises a very valid and interesting point. To further examine whether the reduction in spine density is due to spine elimination or, conversely, the reduction of new spine formation, we re-analyzed our data to quantify the percentage of spines eliminated and formed 24 h post-axotomy. Interestingly, our data show that *both* an increase in elimination and decrease in new formation occurred. Given the short deadline, we were unable to undertake time-lapses of filopodia with sufficient time resolution to know the fate and transition of each spine. Nonetheless, our finding that both elimination and new formation of spines are affected following axotomy is a finding that greatly strengthens this manuscript. This new data is shown in **Figure 2b-d** and discussed in the text [Results, section entitled 'Spine density decreases after distal axon injury', 2nd paragraph, page 6].

3) Page 7: "Our results suggest that distal axotomy triggers a retrograde cascade leading to enhanced presynaptic excitability". I agree only partially with this statements. Data can be interpreted slightly differently: if the spine loss would occur mainly to silent synapses, we would still see a net increase of excitability through a post synaptic mechanisms of selective spine elimination. Indeed, the increase in size of the mEPSC showed in Fig 2h can be due either to the selective elimination of silent or quasi-silent spines, or to the selective enrichment of AMPAR of the remaining spines. To summarize, figure 2 indicates that both pre and post-synaptic mechanisms are into play.

The reviewer is correct that both post-synaptic and presynaptic changes are likely in play to modulate excitability. We believe that our FM data conclusively indicates a presynaptic enhancement in excitability, but that the enhancement in mEPSC frequency may have both presynaptic and postsynaptic mediators. Interestingly, we added more data to increase the sample size of mEPSC recordings and found that the mEPSC amplitude difference between axotomized and uninjured neurons was no longer significant ($p = 0.20$) [Figure 3g; Results, section entitled 'Enhanced glutamate release occurs at synapses onto injured neurons', pages 8-9].

4) It would be very interesting to perform the immunohistochemistry for PSD95 to see on an individual spine basis whether the excitability increase would come with an enhancement of the post-synaptic density. On a similar note, it would be interesting to see the statistics of the spine head volume before and after axotomy. This can be estimated by the integral of the spine head fluorescence normalized for the nearby dendrite fluorescence. Depending on the characteristics of the imaging data acquired for the study, all the necessary data might already be there and would "only" require proper analysis.

As described in comment 3 above, we no longer found that the mEPSC current amplitudes were significantly different than in uninjured controls when we increased our sample sizes. As such, we believe these additional experiments are not within the scope of the manuscript.

5) The data about GABAergic synapses is quite interesting and the immune data are pretty convincing. Here I have a very basic issue that should somehow be mentioned. At the time point of the study (13 DIV) the intracellular Cl concentration should still be immature in a fraction of neurons, therefore, GABA currents might be depolarizing. As I mentioned before (point 2), what might be happening here might not be completely representative of what would happen in the mature brain. This point would be put to rest with the analysis of mIPSC (similarly to what done for mEPSC) by means of perforated patch (so that not to alter the Cl content of the cell). These experiments would also allow to estimate the Cl reversal, which could be quite interesting...

We thank the reviewer for this suggestion and agree that alterations in intracellular chloride concentration may influence the effects of axotomy on GABAergic transmission at more mature developmental time points. As the reviewer suggested, we addressed this concern by recording mIPSCs from axotomized neurons, their neighboring uninjured neurons, and control neurons from separate, completely uninjured microfluidic chambers. These new data are shown in Figure 5d-f. To approximate conditions that exist in brain during later development, we recorded from these neurons using an intracellular solution that contained a low (10 mM) chloride concentration that is approximately the same concentration that

exists in mature neurons (Owens et al. 1996; Yamada et al. 2004). In our conditions, mIPSCs were hyperpolarizing currents, just as GABAergic transmission typically is in the mature brain. Therefore, our results demonstrating an increased frequency of spontaneous GABAergic transmission following axotomy are likely generalizable to the intracellular chloride conditions that would exist in the mature brain. These data are now described in Results, section entitled 'Axotomy selectively eliminates GABAergic terminals onto spines of injured neurons', 2nd paragraph, pages 11-12 and the experimental conditions in Methods, section 'mIPSC recordings', page 31.

6) One missing aspect of this study is the lack of a mechanistic interpretation between the axotomy and the observed changes. Axotomy is known to activate several signaling mechanisms (well described by Rishal and Fainzilber, 2014) occurring at different time points after the lesion. The earliest signaling mechanism, compatible with the timing of the transcription-block experiments, include a retrograde propagating calcium wave which depends on both the activation of NaV channels and on the reversed operation of the NaCa exchange. Surprisingly, the TTX experiments seem to rule out the involvement of retrograde firing on signaling from the lesion site to the cell bodies. However, depending on the details of the axotomy process, the exchanger could be inverted if enough Na would leak in at the lesion site. This could still trigger the formation of a Ca wave that would back-propagate along the axon. It would be interesting to observe whether the blockage of the NaCa exchange at the lesion time (with Bepredil, for example) would have an effect on the response to axotomy.

We thank the reviewer for raising these important points and for suggesting this experiment. We have now added more mechanistic detail and new data showing that when axotomy is performed in the presence of locally applied activity blockade solution (TTX, low calcium, and high magnesium) to the axonal compartment for 1 h during axotomy, that spine loss is prevented 24 h after axotomy (new **Figure 6a,b**). This data suggests that sodium channel activation and/or calcium influx *at the site of injury* triggers retrograde signaling and spine loss. This data together with our data showing that blocking transcription also prevents axotomy-induced spine loss and presynaptic release changes, provides an important mechanistic link between axotomy and synaptic changes. [Results, section entitled 'Local activity and differential gene expression regulate axotomy induced synaptic changes', page 13]

In the previous TTX experiment referred to by the reviewer (now **Figure 6 g,h**), we applied TTX solely to the somatodendritic compartment during axotomy for 1 h. These results show that brief activity blockade within the somatic compartment during axotomy did not prevent axotomy-induced presynaptic release changes 48 h post-axotomy. Since TTX was excluded from the axonal compartment, this experiment did not prevent sodium influx at the site of injury.

Minor points:

1) The axotomy process is of obvious crucial importance in this study. Therefore some details should be provided in the Methods even if the technique has been described in details elsewhere.

We have now corrected this and described briefly the axotomy process in the methods in addition to providing citations. This section now states, "Axotomy was performed between 11

and 15 days in vitro (DIV) according to previously published procedures (Taylor et al. 2005; Taylor et al. 2009). Briefly, media was first removed from the axonal compartment and stored for future use. The axonal compartment was then aspirated until completely devoid of fluid. The stored culture media was then returned immediately to the axonal compartment for the duration of the culture time. Microfluidic devices with equivalent viable cell populations were randomly chosen for either axotomy or uninjured control groups.” [Methods, section ‘Microfluidic chambers’, page 21]

2) Figure 1d. The quality of the spine images is poor. It might well be a problem of the PDF I have at hand. It might be useful to process the image with a non-linear stretching in order to show properly the fainter details (like thin spines) without oversaturating the larger spine heads. The use of a non linear stretching should be disclosed in the figure legend.

We agree with the reviewer that these images could be improved. We selected different representative images and inverted the fluorescence image such that one can now clearly see different spine shapes including thin spines (now **Figure 1c**). We also added new images showing dendritic spines using this same approach in new **Figures 2, 6, and 8**.

3) Figure 2g. Panels h and i are far more informative and so panel g feels a bit pleonastic.

We agree with the reviewer and have removed this panel.

4) There are a few repetitions in the Method section that requires some careful reading and editing. In general, I found that the Methods are really a bit too succinct.

We added more detail and tried to eliminate repetitions in the Methods section. We also added more methodological details to describe the additional experiments we performed as part of this resubmission.

Reviewer #3 (Remarks to the Author):

In this paper, Nagendran and colleagues examined the impact of axotomy on the innervation of damaged neurons using an in vitro model system based on the use of microfluid chambers. This system allows the compartmentalization of the end of axons (a compartment where they can be axotomised), the axon shaft and the somatodendritic compartment where changes in synapses onto the injured and uninjured neurons could be monitored. Specific dyes or viral infection on the axonal compartment allows the specific labelling and therefore identification of injured neurons (cell bodies and dendrites) on the somatodendritic compartment. This is a nice system of identifying retrograde and trans-synaptic changes.

Like previously reported, the authors showed that axotomy leads to the loss of dendritic spines on injured neurons. This is also accompanied by the loss of GABA inhibition as determined by the decrease in the number of vGAT puncta in association with remaining spines. Indeed, the loss of GABA inhibition has been proposed to lead to increased

excitability. The authors examined here the mechanisms that contribute to this loss of inhibition by showing that Netrin is involved.

To evaluate presynaptic changes FM dyes were used. The authors found an increase in the unloading rate of FM suggesting increased release onto the dendrites of injured cells. The authors also report that patch clamp recordings of injured neurons reveal an increase in the frequency and amplitude of mEPSCs. The increased frequency is consistent with their finding with FM dye. The authors proposed that axotomy leads to the loss of vGat puncta onto a subset of dendritic spines resulting in increased excitability. To explain the increased FM labelling, the authors proposed that a trans-synaptic signal (Netrin) is involved. However, some of the results do not seem to come together.

1. The authors made a strong case that trans-synaptic signalling from the injured neuron results in a decreased inhibition by GABA, due to the loss of GABA inputs, results in increased excitability. However, recordings of mIPSCs are missing. Importantly, many of quantifications for FM puncta or innervation (vGluT and vGat puncta) were normalized so it is difficult to assess the effect.

We thank the reviewer for raising these important points. We have now added the analysis of mIPSC recordings 48h post-axotomy to our manuscript. Interestingly, we found an enhancement in mIPSC frequency at axotomized neurons 48 h post-axotomy. While inhibitory terminals onto axotomized neurons were reduced at this time point, the remaining inhibitory terminals appear more spontaneously active, suggesting a compensatory scaling effect. We also examined whether this increased rate of spontaneous GABA release was restricted to directly injured neurons. Within the axotomized cultures, we compared both cut and uncut neurons and found that the mIPSC frequency was increased in both groups, but was not different between the directly axotomized neurons and their uncut neighbors. This suggests that the alteration of inhibitory synaptic transmission following axotomy affects both directly injured and neighboring, uninjured neurons. These new data are shown in **Figure 5d-f** and described in Results, section entitled 'Axotomy selectively eliminates GABAergic terminals onto spines of injured neurons', 2nd paragraph, page 11-12.

We also performed additional experiments to confirm the loss of inhibitory terminals on axotomized neurons using an anti-vGAT antibody. For new **Figure 5c** we have now presented these data as vGLUT1 puncta per μm^2 and vGAT puncta per μm^2 (not normalized). These data show the same trends as in **Figures 5a,b** which quantified the fraction of vGLUT1+ and GAD67+ FM puncta. These data are described in Results, section entitled 'Axotomy selectively eliminates GABAergic terminals onto spines of injured neurons', 1st paragraph, page 11.

2. The authors also showed that TTX, which blocks action potentials, does not affect the effects induced by axotomy. TTX should affect both excitatory and inhibitory synapses. One would presume that TTX would affect inhibitory inputs and therefore block some of the effect of axotomy. The results are not explained.

We apologize for our lack of clarity in describing this experiment. We added TTX to the somatic compartment briefly for 1 h during axotomy and then measured FM unloading rate 48 h post-axotomy. Because of this brief treatment time we would not expect for TTX to have a persistent effect on excitatory or inhibitory synapses. The text and figure legend has been

updated to make these details clearer [**Figure 6f,g**; Results, section entitled 'Local activity and differential gene expression regulate axotomy induced synaptic changes', 2nd paragraph, page 14].

3. Finally, the microarray data is poor. The authors showed two figures with just a number of dots representing the different genes regulated following axotomy but they only labelled one (netrin). They only mentioned that Netrin is affected after axotomy as data not shown (??). Importantly, the table containing the list of genes identified in their screen is missing! The table only has the title.

We have now added a supplemental figure showing quality control data for our microarrays (**Figure supplement 6**). We thought that we had added the table correctly containing the list of genes, but obviously it wasn't. We have now made the table into a PDF instead of an excel file to ensure that it will be properly uploaded. In our microarray data set, netrin-1 is significantly downregulated. The "data not shown" statement was in reference to our RNA-seq data which also showed netrin-1 downregulated. Since there was only one replicate of this RNA-seq data, we have chosen for this resubmission to eliminate the inclusion of this data. Importantly, we performed a new analysis of data gathered from an independent laboratory which performed a microarray study of cortical layer V/VI of rats following spinal cord hemi-transection in which many of these corticospinal neurons were injured. By 7 days following injury, netrin-1 mRNA was significantly downregulated *in vivo* (new **Figure 7**). In addition, other independently published studies have shown that netrin-1 signaling is downregulated following SCI in adult rats, further supporting our selection of netrin-1 as a potential candidate mediating synaptic remodeling. Because of this additional data, we removed the previous scatter plot of cell-cell adhesion transcripts and instead supply this data as **Table Supplement 3**. Netrin-1 was the only transcript within the GO cell-cell adhesion category significantly downregulated in both the *in vitro* and *in vivo* samples; this suggests a reliable response considering the different samples and collection conditions. This new *in vivo* analysis and relevant studies are described in Results, section entitled 'Netrin-1 mRNA down-regulated post-axotomy', page 15.

4. In Figure 1, the authors showed that processes grow from the somato-dendritic compartment into the grooves. However, the authors did not demonstrate that these processes are axons. Proper staining for axonal and dendritic markers should be used to demonstrate the specificity of the compartmentalization.

We added this data in **Figure supplement 1a**. In addition, we have characterized this axonal isolation using the same cell-type in previous studies which we have now referenced (Taylor et al. 2005; Taylor et al. 2009; Taylor et al. 2010).

5. Figure 1d: The data on spine density should not be normalized to 1 but presented as spines per unit length of dendrite.

This has now been corrected (new **Figure 1c**). In addition, we now show spines per unit length of dendrite for **Figures 2, 6, 8**.

6. Figure 1d: the data on synapse density after 24 hour axotomy does not coincide with the data presented in Figure suppl 2. Here there is a 25% decrease, in supplementary figure 2 there is a 70% decrease. The authors need to explain this discrepancy.

We thank the reviewer for raising this important point. **Figure supplement 2** compares neurons that have been fixed from different chambers axotomized and uninjured at 14 DIV, 24 h after axotomy. New **Figure 1c** shows spine density before (at 13 DIV) and after injury (14 or 15 DIV) from the same neurons. Spine density in the uninjured controls increases gradually as cultures mature at these ages; thus, in the fixed samples there is expected to be a greater difference in spine density because the control was imaged at 14 DIV. This clarification is now explained in the figure legend of **Figure supplement 2** where we state, “A greater reduction in spine density was quantified in these fixed samples compared to the before and after axotomy data shown in Figures 1 and 2; this can be explained by the age of imaging as spine density gradually increases over time in the control uninjured cultures.”

7. Figure 2 and page 7, first paragraph. The authors indicated that after 48 hour axotomy there is an increase in the number of responsive FM puncta although the total number does not change. The conclusion is that there is shift from silent to responsive FM puncta. This finding poses a question. If the number of spines decreases but the total number of FM puncta onto the injured cell does not, then are some of these puncta orphan sites? If so, how do they contribute to excitability?

We have reworded this section to clarify our data. We found that both spine density and the total number of FM puncta were reduced 48 h after axotomy (**Figures 1c,d** and **3e**, respectively). Even though there were fewer FM puncta, we found that this smaller pool of FM puncta had a higher fraction of responsive puncta 48 h post-axotomy. Because our IF data showed a specific and significant decrease in both GAD67-positive FM puncta and vGAT-labeled synapses colocalized to axotomized neurons (**Figure 5a-c**), we believe that the reduction in FM puncta at 48 h post-axotomy reflects the absence of these inhibitory terminals at this time point. Our data does suggest that excitatory terminals might be preferentially spared compared with inhibitory terminals (**Figure 5a-c**). New **Figure 5c** shows that vGLUT1 puncta colocalized with axotomized neurons are maintained at a similar density in uninjured controls and 48 h post-axotomy. These spared excitatory terminals could form shaft synapses or form multiple excitatory presynaptic inputs onto individual spines. Alternatively, some of these terminals may be orphan presynaptic sites. The increased spontaneous release of glutamate at injured neurons 48 h following axotomy, without an increase in the number of excitatory terminals, suggests that the maintained excitatory inputs have an increased release rate after injury and may contribute to the hyper-excitability post-injury. These points are now discussed in the Discussion, 3rd paragraph, page 18.

8. In Figure 2, it is not clear whether the FM puncta that change are those in apposition to the injured cell. This is crucial to determine how specific the labeling using the FM dye is.

-it appears that puncta within the vicinity of the injured neurons show increased release rate.

The reviewer raises a good point. First, we examined the colocalization of the FM dye labeling with a presynaptic marker, synapsin I, and found 93% colocalization, indicating that the vast majority of terminals were labeled with FM dye (**Figure supplement 3**). Second, we performed new experiments to test the specificity of FM unloading at synapses formed onto directly injured neurons compared with FM unloading at uninjured neurons within the same cultures. To do this, we used axotomized chambers which included a population of axotomized neurons as well as uninjured neurons that did not extend axons into the axonal

compartment. We identified axotomized versus uninjured neurons by either the presence or absence of eGFP labeling introduced via a modified rabies virus added to the axonal compartment. As we expected we found that these remote uninjured regions had a significant decrease in FM unloading compared with directly injured neurons. Further, the unloading curves of these remote uninjured regions were indistinguishable from unloading curves of uninjured control neurons labeled with eGFP rabies virus. These new data are shown in **Figure 3i** and described in Results, Section entitled 'Enhanced glutamate release occurs at synapses onto injured neurons, 2nd paragraph, page 9.

9. Figure 3 shows normalized data on the fraction of vglut/spines. This data, like in other figures, should be shown as number of vglut puncta along a given length of dendrite.

We have now added new data quantifying vGLUT1 puncta per area in new **Figure 5c** (new Figure 5 replaces old Figure 3). We chose to quantify puncta per area—the area corresponding to the axotomized (or uninjured) neuron retrograde labeled using a modified rabies virus as described previously. For new **Figure 5c** we have now presented these data as vGLUT1 puncta per μm^2 and vGAT puncta per μm^2 (not normalized). For new **Figure 5g-l** we quantified the fraction of vGLUT1-positive and vGAT-positive spines; these data are no longer normalized. In addition, for **Figure 8e,f,j** we have presented these data as number of puncta per μm^2 (not normalized). The neuron cultures used for the data presented in **Figure 8j** were more mature than the cultures used in **Figure 8e,f** (14DIV); thus, overall fractions of vGLUT1 and vGAT puncta were slightly higher in **Figure 8j** because of this added maturation time.

10. Figure 4. The authors used TTX to examine its impact on the response after axotomy and reported that TTX does not affect the axotomy-mediated response on FM. This is puzzling because TTX should affect both inhibitory and excitatory synapses. So inhibitory terminals should be silenced. A possible conclusion is that the loss of inhibitory inputs is not relevant to increase presynaptic response. The authors should reconcile these apparently conflicting results.

Please refer to our response to your comment #2. TTX was applied to the somatic compartment for 1 h during axotomy. We quantified inhibitory and excitatory synaptic responses 48h post-axotomy. We apologize for our lack of clarity and have tried to describe our experimental timeline more clearly.

11. Figure 5, it is surprising that the authors did not reveal the identity of the genes in this graph other than netrin.

We apologize for this oversight and have now added the identities of the differentially expressed cell-cell adhesion transcripts as **Table supplement 3**. We removed the previous volcano plot of these transcript and replaced it with data showing netrin-1 gene expression is downregulated *in vivo* in a nerve injury model (**Figure 7c**).

12. Table Supplement 2, has a title that says "List of transcripts that are significantly changed 24 h after injury". However, the table is empty. Any explanation?

We apologize for this error in uploading our data. We are unclear how this happened. We have now generated a PDF table of the differentially expressed transcripts (instead of an

excel file which we uploaded for the first submission). We have now uploaded all our microarray data to GEO.

13. Figure 6, the number of DCC puncta in association with the injured neuron and also on non-injured neuron should be presented.

The DCC immunofluorescence was too diffuse, especially in the axotomy condition, to accurately assess puncta numbers. Instead, we show mean fluorescence intensity per spine ROI. This is included in **Figure 8g,h**.

14. Figure 6g seems to show that Netrin rescues the decrease in GAD67 puncta after axotomy but there is a clear tendency to decrease. The analysis was done on 7 chambers. Given that no details are given about how many experiments these 7 chambers come from, the authors should provide clear evidence that this effect or lack of effect is real. For example, they should also analyse the number of vGAT puncta.

We added an additional experiment quantifying the number of vGAT puncta per area of injured neuron. These data show the same trend as the GAD67 data, where we see approximately a 50% reduction in inhibitory puncta following axotomy. These data are shown together in new **Figure 8**. We also found that netrin-1 treatment following axotomy increased the number of inhibitory puncta to control levels, adding support that netrin-1 signaling influences axotomy-induced synaptic remodeling.

- Ahn, K. J., I. A. Seo, H. K. Lee, E. J. Choi, E. H. Seo, H. J. Lee and H. T. Park (2007). "Down-regulation of UNC5 homologue expression after the spinal cord injury in the adult rat." Neuroscience letters **419**(1): 43-48.
- Banker, G. and K. Goslin (1998). Culturing Nerve Cells. Cambridge, MIT Press.
- Banker, G. A. and W. M. Cowan (1977). "Rat hippocampal neurons in dispersed cell culture." Brain research **126**(3): 397-342.
- Goldman, J. S., M. A. Ashour, M. H. Magdesian, N. X. Tritsch, S. N. Harris, N. Christofi, R. Chemali, Y. E. Stern, G. Thompson-Steckel, P. Gris, S. D. Glasgow, P. Grutter, J. F. Bouchard, E. S. Ruthazer, D. Stellwagen and T. E. Kennedy (2013). "Netrin-1 promotes excitatory synaptogenesis between cortical neurons by initiating synapse assembly." The Journal of neuroscience : the official journal of the Society for Neuroscience **33**(44): 17278-17289.
- Han, X., Y. Zhang, L. Xiong, Y. Xu, P. Zhang, Q. Xia, T. Wang and Y. Ba (2016). "Lentiviral-Mediated Netrin-1 Overexpression Improves Motor and Sensory Functions in SCT Rats Associated with SYP and GAP-43 Expressions." Molecular neurobiology.
- Lu, H., Y. Wang, F. Yuan, J. Liu, L. Zeng and G. Y. Yang (2011). "Overexpression of netrin-1 improves neurological outcomes in mice following transient middle cerebral artery occlusion." Frontiers of medicine **5**(1): 86-93.
- Manitt, C., D. Wang, T. E. Kennedy and D. R. Howland (2006). "Positioned to inhibit: netrin-1 and netrin receptor expression after spinal cord injury." Journal of neuroscience research **84**(8): 1808-1820.
- Menon, S., N. P. Boyer, C. C. Winkle, L. M. McClain, C. C. Hanlin, D. Pandey, S. Rothenfusser, A. M. Taylor and S. L. Gupton (2015). "The E3 Ubiquitin Ligase TRIM9 Is a Filopodia Off Switch Required for Netrin-Dependent Axon Guidance." Developmental cell **35**(6): 698-712.

- Owens, D. F., L. H. Boyce, M. B. Davis and A. R. Kriegstein (1996). "Excitatory GABA responses in embryonic and neonatal cortical slices demonstrated by gramicidin perforated-patch recordings and calcium imaging." The Journal of neuroscience : the official journal of the Society for Neuroscience **16**(20): 6414-6423.
- Sun, H., T. Le, T. T. Chang, A. Habib, S. Wu, F. Shen, W. L. Young, H. Su and J. Liu (2011). "AAV-mediated netrin-1 overexpression increases peri-infarct blood vessel density and improves motor function recovery after experimental stroke." Neurobiology of disease **44**(1): 73-83.
- Taylor, A. M., N. C. Berchtold, V. M. Perreau, C. H. Tu, N. Li Jeon and C. W. Cotman (2009). "Axonal mRNA in uninjured and regenerating cortical mammalian axons." J Neurosci **29**(15): 4697-4707.
- Taylor, A. M., M. Blurton-Jones, S. W. Rhee, D. H. Cribbs, C. W. Cotman and N. L. Jeon (2005). "A microfluidic culture platform for CNS axonal injury, regeneration and transport." Nat Methods **2**(8): 599-605.
- Taylor, A. M., D. C. Dieterich, H. T. Ito, S. A. Kim and E. M. Schuman (2010). "Microfluidic local perfusion chambers for the visualization and manipulation of synapses." Neuron **66**(1): 57-68.
- Yamada, J., A. Okabe, H. Toyoda, W. Kilb, H. J. Luhmann and A. Fukuda (2004). "Cl⁻ uptake promoting depolarizing GABA actions in immature rat neocortical neurones is mediated by NKCC1." The Journal of physiology **557**(Pt 3): 829-841.

Reviewers' comments:

Reviewer #1 (Remarks to the Author):

The authors have addressed most of the issues I had previously raised. The manuscript is greatly improved. I have few remaining comments.

The most important relates to the numbers presented in several of the figures (eg. 1-2-3-5-8), which seem low (e.g. in Fig 1, 10 neurons in total (from 2 independent experiments) is a very low number or in Fig 8f, 3-4 neurons per condition is also a low number). Have the authors done any power calculations to determine the n required to obtain appropriate statistical power (i.e. 0.8, Cohen 1988)?

How many neurons/spines were considered in each experiment? How many different chambers were analysed in each experiment? This information should always be included in the figure legend, but is not always reported for each experiment. E.g.

Figure 1: What were the total number of spines counted? And the total dendritic length? Without this information it is difficult to establish the strength of the findings.

Supplementary Fig 1c-d-e: how many chambers were analysed?

Supplementary Fig2b: how many chambers were analysed?

The other main comment relates to the blinding, or lack of it. The authors say: 'Blinding during group allocation was not feasible because neurons were followed before and after axotomy.' However this is not acceptable as the data can be analysed by an independent analyser (not the person who ran the experiment) who is blind to experimental condition. This is very important.

Few typos were spotted:

Line 100: vivo..

Line 884: Astericks

Reviewer #2 (Remarks to the Author):

The AA provided interesting new data and responded to all the points raised in my review.

Reviewer #3 (Remarks to the Author):

The authors have made a significant effort in answering my comments and made substantial improvement to the paper by including a large set of new data. However, the new results increase the complexity of the conclusions which need to be addressed more clearly. In addition, the authors need to address a question regarding the quality of one of the images.

The authors have added new results by measuring mIPSCs in neurons 48 hours after axotomy. The authors showed that axotomy increases the frequency of mIPSCs even though the number of GABA (GAD67 positive) terminals on dendritic spines decreases. This suggests an increase in GABA release. One would presume that these changes in mIPSC frequency could decrease excitability. This should be more clearly discussed in the discussion section of the paper.

In Figure 8 the authors examined the levels of DCC, the receptor for Netrin, after axotomy. They provide lower magnification images of the levels of DCC on axotomised neurons (Figure 8G). However this data is not convincing because the selected area was taken from a part of the culture where there is an overall lower level of DCC. More convincing data should be presented.

Minor point:

In point 2 of the rebuttal letter, the authors indicated that they have now provided data from the array experiments in Figure supplementary 6. However, this figure shows fluorescence micrograph images. The data is in Table Supplement 3 as indicated later in the letter.

Reviewer #1 (reviewer comments in bold):

The authors have addressed most of the issues I had previously raised. The manuscript is greatly improved. I have few remaining comments.

1. The most important relates to the numbers presented in several of the figures (eg. 1-2-3-5-8), which seem low (e.g. in Fig 1, 10 neurons in total (from 2 independent experiments) is a very low number or in Fig 8f, 3-4 neurons per condition is also a low number). Have the authors done any power calculations to determine the n required to obtain appropriate statistical power (i.e. 0.8, Cohen 1988)?

We thank the reviewer for bringing up these important points. To address the reviewer's concern we focused the last 3 months on increasing the number of experimental replicates. In addition, we performed statistical power calculations for our experiments and have now established that we have sufficient statistical power of at least 0.8 and using a 5% error rate (type I). Justification for our sample size is provided in the Methods section under statistics and is copied here:

“For the paired spine density analyses (Fig. 1d, 6b-c, 8b, 8i), we calculated a minimum sample size of 10 primary dendritic processes per condition using means and standard deviations from Fig. 1d. For FM unloading analyses (Fig. 3c, 3i, 6e, 6g, suppl. 3b-f), we used a minimum sample size of 50 FM puncta per condition based on previously published data²⁴ and using means and standard deviations from Fig. 3c. For unresponsive/responsive FM puncta analyses (Fig. 3d-e, 6d, 6f, 8c-d), we calculated a minimum sample size of 6 frames per condition using means and standard deviations of unresponsive puncta in uninjured control versus 48h post-axotomy (Fig. 3e). For vGAT/vGLUT1 puncta per area analyses (Fig. 5c, 8e-f, 8j), we calculated a minimum sample size of 8 neuron fields per condition using the means and standard deviations of vGAT puncta per area in control and axotomized samples (Fig. 5c). For the fraction of GAD67 or vGLUT1 puncta to FM puncta analyses (Fig. 5a,b), we calculated a minimum sample size of 18 frames per condition using the means and standard deviations found in Fig. 5b. For DCC immunofluorescence per spine analyses (Fig. 8h), we calculated a minimum of 75 samples per condition using the means and standard deviations of controls and axotomized samples. We estimated requisite samples size for recordings of spontaneous postsynaptic currents by assuming a doubling or halving of mEPSC or mIPSC parameters following axotomy in correlation with the magnitude of changes observed in our analysis of spine numbers, presynaptic release data, and based on similar effect sizes being reported following synaptic activity blockade^{63,64}. Standard deviation values for the power analysis were estimated using previously published values from dissociated hippocampal neuron electrophysiological recordings (Henry et al, 2012 and Hartman et al, 2006). This analysis suggested we required 13 samples per group for mEPSCs and 9 samples per group for mIPSCs. All recording, (Figures 3g-h, 5d-f) meet or exceed the sample size required by our power analysis. For *in vivo* experiments, hypotheses regarding spike firing rates were tested independently in each cortical layer (Va, Vb and VI) using a two-tailed t-test, ($\alpha = 0.05$). Samples were not excluded from our data sets and sample size was determined based on our experience and previous publications²⁴. As confirmation of sufficient sample sizes, we used the means and standard deviations for layer Vb control and SCI conditions and calculated a minimum of 112 samples for control and 282 samples for SCI conditions. For all experiments, sample sizes were equal or greater than the calculated minimum sample sizes.”

2. How many neurons/spines were considered in each experiment? How many different chambers were analysed in each experiment? This information should always be included in the figure legend, but is not always reported for each experiment. E.g.

Figure 1: What were the total number of spines counted? And the total dendritic length? Without this information it is difficult to establish the strength of the findings.

Supplementary Fig 1c-d-e: how many chambers were analysed?

Supplementary Fig2b: how many chambers were analysed?

We now include the numbers of neurons, total dendritic length, and spines considered within the figure legends. We refer the reviewer to these legends, especially Figs. 1, 2, 6, and 8. Note that we edited all legends to <350 words according to the journal guidelines. We also include the numbers of chambers and experimental replicates. After much consideration, we choose to eliminate Supplementary Fig. 2, which quantified dendritic spines in fixed uninjured control and axotomy conditions. We did so because the rigorous live imaging experiments in Figs. 1 & 2 largely replicate and supersede the data in the supplementary figure we eliminated. Additional replicates and blinding further validate and increase the strength of our findings.

3. The other main comment relates to the blinding, or lack of it. The authors say: ‘Blinding during group allocation was not feasible because neurons were followed before and after axotomy.’ However this is not acceptable as the data can be analysed by an independent analyser (not the person who ran the experiment) who is blind to experimental condition. This is very important.

We took additional time to perform our analyses blinded as appropriately suggested by the reviewer. We thank the reviewer for this suggestion that greatly strengthens confidence in our findings.

4. Few typos were spotted:

Line 100: vivo..

Line 884: Astericks

These typos have now been fixed.

Reviewer #2 (Remarks to the Author):

The AA provided interesting new data and responded to all the points raised in my review.

We thank the reviewer for the encouraging response.

Reviewer #3 (Remarks to the Author):

The authors have made a significant effort in answering my comments and made substantial improvement to the paper by including a large set of new data. However, the new results increase the complexity of the conclusions which need to be addressed more clearly. In addition, the authors need

to address a question regarding the quality of one of the images.

1. The authors have added new results by measuring mIPSCs in neurons 48 hours after axotomy. The authors showed that axotomy increases the frequency of mIPSCs even though the number of GABA (GAD67 positive) terminals on dendritic spines decreases. This suggests an increase in GABA release. One would presume that these changes in mIPSC frequency could decrease excitability. This should be more clearly discussed in the discussion section of the paper.

The reviewer raises an important point. At first glance, one might expect that mIPSC frequency would decrease following axotomy because of the net increase in excitability. However, we found the opposite. mIPSC frequency significantly increases 48h following axotomy. A likely explanation is that the increased rate of GABA release from remaining inhibitory terminals could be compensating for the hyper-excitability and loss of inhibition at the injured neuron. This point is now described in the Discussion at the end of the 4th paragraph.

2. In Figure 8 the authors examined the levels of DCC, the receptor for Netrin, after axotomy. They provide lower magnification images of the levels of DCC on axotomised neurons (Figure 8G). However this data is not convincing because the selected area was taken from a part of the culture where there is an overall lower level of DCC. More convincing data should be presented.

We have now used representative images that contain roughly equivalent overall levels of DCC immunofluorescence throughout the frame (Fig. 8g). We found greater clustering of DCC at dendrites and synapses in uninjured control and axotomized+netrin-1 compared with axotomized alone conditions.

3. Minor point: In point 2 of the rebuttal letter, the authors indicated that they have now provided data from the array experiments in Figure supplementary 6. However, this figure shows fluorescence micrograph images. The data is in Table Supplement 3 as indicated later in the letter.

These issues have been corrected. Thank you.

REVIEWERS' COMMENTS:

Reviewer #1 (Remarks to the Author):

The authors addressed all my comments. I think the study is much improved and stronger now and i'm happy for it to be published.

Reviewer #3 (Remarks to the Author):

The authors have addressed all the points raised by this reviewer.

I strongly recommend publication in Nature Communication.